# Global stabilization of the transcriptome in mitotic cells

Ekaterina Khalizeva [ID][1,2], Arash Latifkar[1,2,3], Kehui Xiang[1,2,3], David P Bartel [ID][1,2,3], Jimmy Ly [ID][1,2]✉ & Iain M Cheeseman [ID][1,2]✉

## Abstract

In the presence of cell division errors, mammalian cells can pause in mitosis for tens of hours with little to no transcription, while still requiring continued translation for viability. These unique aspects of mitosis require substantial adaptations to gene expression. During interphase, homeostatic control of mRNA levels involves a constant balance of transcription and degradation, with a median mRNA half-life of ~2–4 h. If such short half-lives persisted in mitosis, cells would be expected to rapidly deplete their transcriptome without new transcription. Here, we report that the transcriptome is globally stabilized during prolonged mitotic delays. Median mRNA half-lives are increased >4-fold during mitotic arrest compared to interphase, buffering mRNA levels in the absence of new synthesis. Moreover, poly(A) tail-length profiles change during mitotic arrest, strongly suggesting a partial mitotic repression of deadenylation. In contrast, siRNA-directed mRNA degradation machinery remains active. We further show that mitotic mRNA stabilization depends on PABPC1&4. Depletion of PABPC1&4 during mitotic arrest reduces mRNA stability and disrupts the cells' ability to maintain arrest, highlighting the critical physiological role of mitotic transcriptome buffering.

**Keywords** Mitosis; mRNA Stability; Gene Regulation; Deadenylation
**Subject Categories** Cell Cycle; Chromatin, Transcription & Genomics; RNA Biology

## Introduction

Cellular mRNA levels are controlled by constant turnover, balancing transcription and mRNA degradation. This homeostatic control promotes remarkable resilience in the presence of various physiological stressors, allowing cells to compensate for perturbations in mRNA synthesis by reducing degradation, and vice versa (Hartenian and Glaunsinger, 2019; Slobodin et al, 2020; Sun et al, 2013; Haimovich et al, 2013). The global regulation of mRNA stability becomes particularly important in contexts in which

mammalian cells attenuate transcription for prolonged periods (Hartenian and Glaunsinger, 2019; Helenius et al, 2011). For example, mouse oocytes have very low levels of RNA synthesis during meiotic maturation, which takes weeks to occur (Brower et al, 1981; Jahn et al, 1976; Moore and Lintern-Moore, 1978). Oocytes resume transcription only after fertilization and zygotic genome activation. During these transcriptionally silent stages of oogenesis and early embryogenesis, maternal mRNA stability is mediated, at least in part, by the germline-specific RNA-binding protein MSY29 (Medvedev et al, 2011) and by subcellular mRNA localization (Cheng et al, 2022). The maintenance of maternal mRNAs is crucial for timely translational activation and proper oocyte maturation during and after the transcriptionally silent stages of development (Conti and Kunitomi, 2024; Winata and Korzh, 2018).

Mitotic cells also exhibit significantly downregulated transcription, accompanied by chromosome condensation (Antonin and Neumann, 2016), eviction of transcriptional machinery from chromatin (Parsons and Spencer, 1997; Martínez-Balbás et al, 1995; Perea-Resa et al, 2020), inactivation of general transcription factors (Segil et al, 1996), and low levels of nascent transcript labeling (Taylor, 1960; Palozola et al, 2017). Although mitosis typically lasts for ~1 h, errors induced by either physiological damage or anti-mitotic drugs can cause somatic cells to arrest in mitosis for hours (Gascoigne and Taylor, 2008; Skoufias et al, 2006). This arrest is mediated by the conserved Spindle Assembly Checkpoint signaling pathway (McAinsh and Kops, 2023). During an extended mitotic delay, ongoing translation is essential for cell survival and the ability to maintain a cell cycle arrest (Sloss et al, 2016; Tsang and Cheeseman, 2023; Ly et al, 2024). Following the resolution of mitotic damage, cells can subsequently segregate their chromosomes, exit mitosis, and re-enter G1, once again becoming transcriptionally competent (Palozola et al, 2017).

In asynchronous mammalian somatic cells, the median mRNA half-life ($t_{1/2}$) is typically 2–4 h, depending on the cell type and the half-life measurement strategy (Eisen et al, 2020; Herzog et al, 2017; Lugowski et al, 2018). This suggests that, during prolonged periods of transcriptional silencing in which ongoing translation is essential, cells may require compensatory mechanisms to prevent depletion of their transcriptomes. However, somatic cells do not express the factors that promote mRNA stabilization in oocytes, such as MSY29, suggesting that there must be distinct strategies for

[1]Whitehead Institute for Biomedical Research, Cambridge, MA, USA. [2]Department of Biology, Massachusetts Institute of Technology, Cambridge, MA, USA. [3]Howard Hughes Medical Institute, Cambridge, MA, USA. ✉E-mail: jimmy1996ly@gmail.com; icheese@wi.mit.edu

regulating mRNA stability. Here, we report that mitotic cells globally stabilize their transcriptomes to maintain an extended mitotic arrest in the near-complete absence of mRNA synthesis. We find that the poly(A) tail-length profile of mRNAs is altered in mitotic cells, indicating partially repressed deadenylation, whereas factors required for siRNA-directed mRNA degradation remain active. In addition, we find that cytoplasmic poly(A)-binding proteins (PABPC1&4) are necessary for mitotic mRNA stabilization. Induction of mRNA degradation in mitosis through the depletion of PABPC1&4 disrupts the cells' ability to maintain prolonged mitotic arrest, highlighting the physiological importance of buffering transcriptome levels in mitosis.

# Results

## The cellular transcriptome is not depleted during prolonged mitotic arrest

To test whether mitotically arrested cells maintain their transcriptome levels despite the prolonged global inhibition of transcription, we first evaluated changes in mRNA levels during an extended mitotic arrest. We synchronized HeLa cells using a double thymidine block followed by treatment with the KIF11 inhibitor S-trityl-L-cysteine (STLC) for 8 h. STLC triggers a mitotic arrest (Fig. EV1A) by activating the Spindle Assembly Checkpoint but does not activate stress response pathways (Ly et al, 2024; Skoufias et al, 2006; Musacchio, 2015). We then performed RNA-seq at 0–1, 6–7, and 24–25 h after entry into mitosis. Normalization to spike-in RNA allowed us to quantitatively compare global mRNA level changes between different timepoints (Fig. 1A). Strikingly, the abundances of most mRNAs remained very close to their initial levels after 6 h, and were only reduced by 32% even after 24 h of mitotic arrest (Figs. 1B,D,E and EV1B; Dataset EV1). The maintenance of transcriptome levels was not due to a resumption of transcription in the arrested cells, as FACS-based analysis to detect the incorporation of the uridine analogue 5-Ethynyluridine (5EU) at the relevant timepoints confirmed a lack of ongoing transcription (Fig. 1C). Similarly, prior work found that actinomycin D-mediated transcriptional inhibition in mitotically arrested cells did not alter mRNA levels (Novais-Cruz et al, 2018; Krenning et al, 2022). Together, these results indicate that cells maintain transcriptome levels throughout a 24-h mitotic arrest, despite a near-complete inhibition of mRNA synthesis, suggesting that a mechanism must be in place to prevent global mRNA depletion.

## mRNA half-lives are increased in mitotic cells

To test whether mRNA decay is perturbed in mitotically arrested cells, we next measured mRNA levels over time following pharmacological inhibition of transcription by actinomycin D in cells arrested in either interphase (G2 stage) or M phase of the cell cycle. For these experiments, we used a double thymidine block followed by an 8-h treatment with the CDK1 inhibitor RO-3306 (G2 arrest) or STLC (mitotic arrest) (Figs. 2A and EV2A). We then calculated mRNA half-lives by fitting abundance values to an exponential curve (Fig. EV2B; Methods). In G2 cells, we observed a median mRNA half-life of 5.6 h, in range with prior measurements of mRNA stability in actinomycin D-treated interphase cells

(Herzog et al, 2017; Lugowski et al, 2018). However, mRNA half-lives were almost fourfold higher in mitotically arrested cells, with a median half-life of 20.1 h (Fig. 2B; Dataset EV2). Notable examples of mRNA stabilization include transcripts such as *MYC*, *KLF10*, and *ATF3*, which are unstable in G2 ($t_{1/2}$ 1.0, 1.4, and 1.9 h, respectively) but underwent little to no apparent degradation in mitotic cells (Fig. 2D). We also observed a similar effect when using the CDK7 inhibitor THZ1 instead of actinomycin D to inhibit RNA polymerase II-dependent transcription (median $t_{1/2}$ of 6.8 h in G2 vs. 33.4 h in M; Fig. EV2C).

We next compared half-lives in G1 vs. M cell cycle stages. For this experiment, cells were synchronized in mitosis by STLC arrest, followed by release from the drug for 1 h to allow cells to exit mitosis and enter G1 (see Methods). Global stabilization was even stronger than when comparing G2 vs. M (median $t_{1/2}$ of 3.1 in G1 vs. 27.3 h in M; Fig. 2C). These results are in line with previously reported waves of mRNA decay upon the M-to-G1 transition (Krenning et al, 2022). Krenning et al (2022) also highlighted several examples of mRNAs that are stable over the course of 2 h of mitotic arrest, which is consistent with our findings of long-term mitotic mRNA stability.

Prior work suggested that there is feedback between translation and mRNA stability in asynchronously growing cells (Jia et al, 2020). Despite a requirement for ongoing protein production, translation is partially attenuated in mitotic cells (Tanenbaum et al, 2015; Ly et al, 2024). Therefore, we considered whether the observed global transcriptome stabilization during mitotic arrest could be explained by the global decrease in translation (Morris et al, 2021; Rosa-Mercado et al, 2024). To test this, we evaluated the degree of stabilization of each transcript relative to its change in translation efficiency between asynchronous and mitotic samples, as measured by ribosome profiling (Ly et al, 2024). However, we did not detect a strong correlation between the changes in mRNA half-life and translation efficiency (Fig. 2E, $R_s = -0.12$). Notably, mRNAs that lacked translational changes during mitosis were still stabilized. We also compared the measured change in stability with the translation status of a given transcript. Long non-coding RNAs (lncRNAs), which typically do not undergo translation, displayed only a modestly lower degree of stabilization during mitotic arrest compared to coding transcripts (3.1- and 3.8-fold changes, respectively; Fig. 2F).

In addition, we evaluated mRNA properties that have been associated with altered stability, such as AU-rich elements (Chen and Shyu, 2011; Agarwal and Kelley, 2022), and P-body localization (Blake et al, 2024). However, none of the features tested showed a strong correlation with the degree of mitotic stabilization (Fig. EV2D–F). These observations are consistent with a global reduction in mRNA degradation that is not dependent on targeted mRNA destabilization factors. Together, these results suggest that there is a transcriptome-wide mRNA stabilization during mitotic arrest, and that a global attenuation of translation cannot explain this behavior.

## siRNA treatment can induce mRNA degradation in mitosis

The global increase in mRNA half-lives that we observed in mitotic cells suggests that mRNA decay is perturbed. mRNA degradation is a multi-step process that involves sequential reactions (Garneau

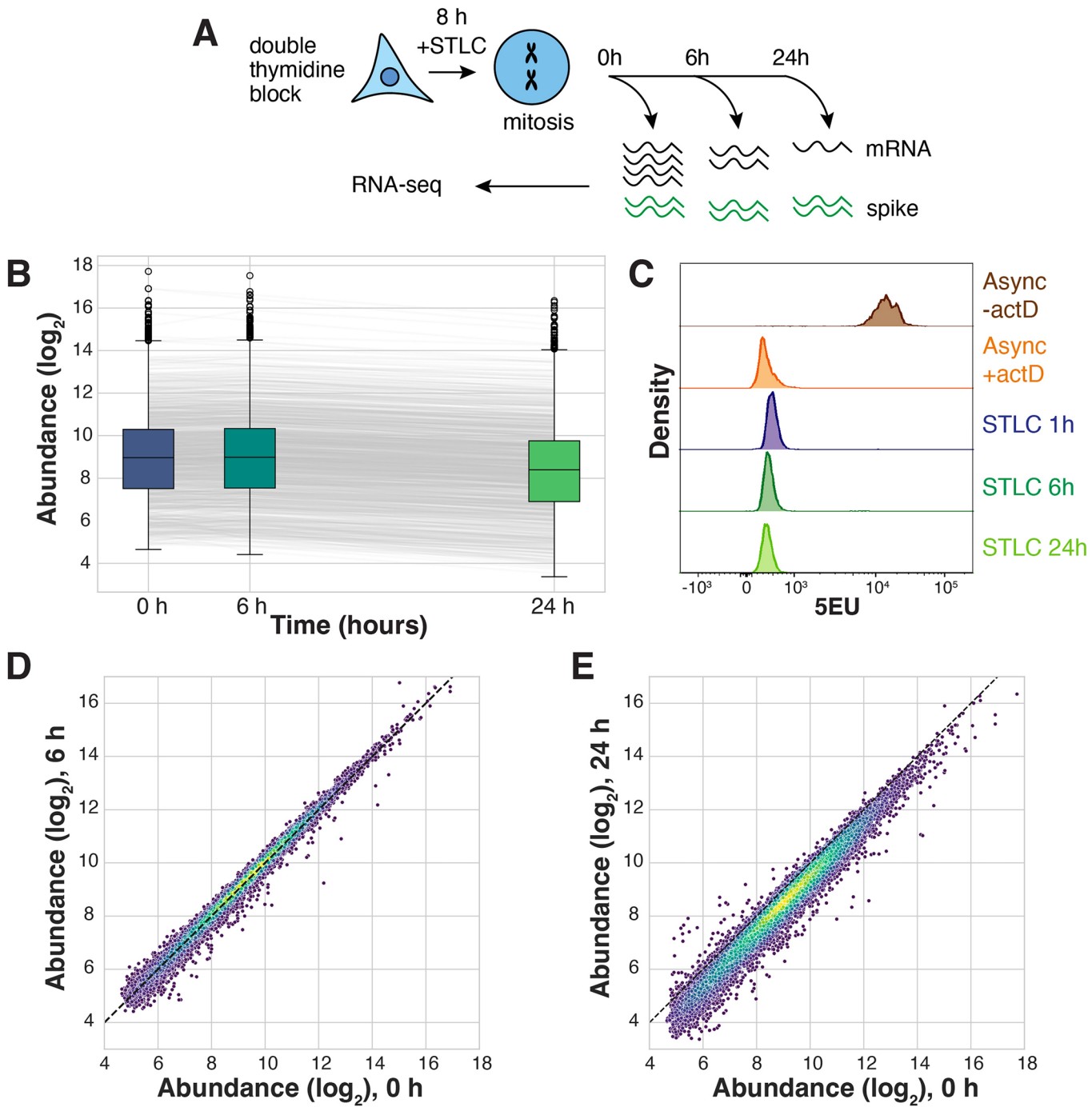

**Figure 1. The cellular transcriptome is not depleted during prolonged mitotic arrest.**

(A) Schematic diagram of timecourse experiments to measure mRNA levels during STLC-induced mitotic arrest. (B) Boxplots comparing spike-normalized mRNA abundances over the course of 24 h of mitotic arrest. Each box shows the interquartile range (IQR) of the data, with the median marked by a horizontal line. The whiskers extend to include the rest of the distribution, except for points that are determined to be outliers based on the IQR. Each line represents the normalized abundance of an mRNA across the timecourse. Each point represents the mean of two biological replicates, $n = 11,282$. (C) Histogram comparing 5EU incorporation in asynchronous samples with or without actinomycin D, and in STLC-arrested samples. (D, E) Scatterplots comparing spike-normalized abundances between (D) 0 and 6 h or (E) 0 and 24 h of arrest. The dotted line indicates the x=y diagonal, points colored by density. Each point represents the mean of two biological replicates, $n = 11,282$.

et al, 2007; Wiederhold and Passmore, 2010; Passmore and Coller, 2022; Dowdle and Lykke-Andersen, 2025). For most mRNAs, decay begins with shortening of the poly(A) tail by cytoplasmic deadenylases (Eisen et al, 2020; Raisch and Valkov, 2022). A subset of deadenylases can displace the poly(A)-binding protein (PABP) and gradually digest the poly(A) tail until it is too short to be bound by a single PABP (<20 nt) (Yi et al, 2018; Webster et al, 2018). After deadenylation, decapping enzymes are recruited to

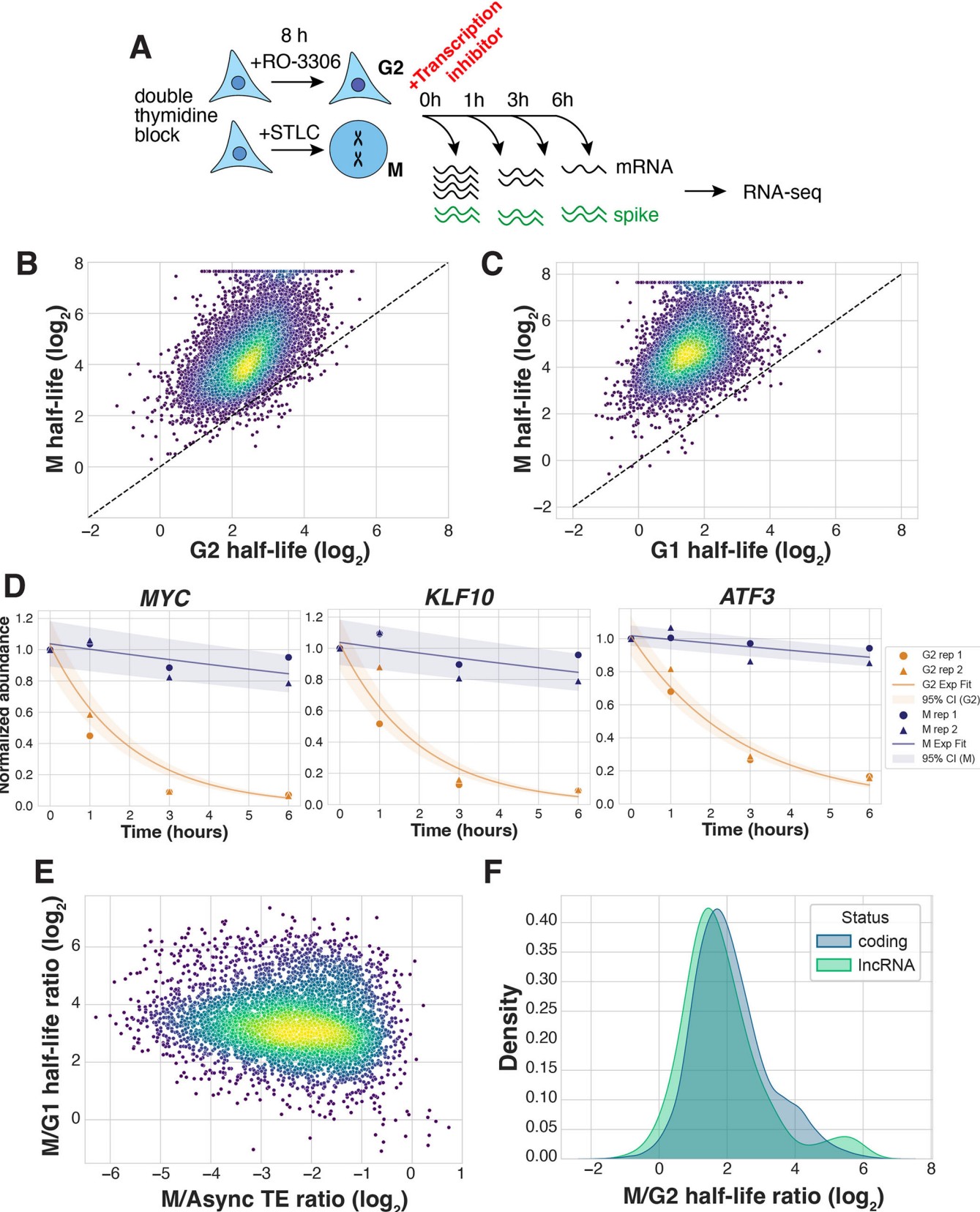

**Figure 2.   mRNA half-lives are increased in cells arrested in mitosis compared to interphase.**

(A) Schematic diagram of transcriptional inhibition timecourse experiments to measure mRNA half-lives in G2- and M-synchronized cells. (B) Scatterplot comparing mRNA half-lives in G2- and M-arrested cells treated with actinomycin D. Each point represents the mean of two biological replicates, $n = 9858$. Median $t_{1/2}$ 5.6 and 20.1 h for G2 and M, respectively. (C) Scatterplot comparing mRNA half-lives in cells synchronized in G1 or M treated with actinomycin D. Data from one biological replicate, $n = 6969$. Median $t_{1/2}$ 3.1 and 27.3 h for G1 and M, respectively. (D) Exponential curve fit to individual transcript abundances, plotted with 95% confidence intervals. *MYC* $t_{1/2}$ 1.0 and 21.2 h, $R^2$ 0.99 and 0.65 in G2 and M, respectively. *KLF10* $t_{1/2}$ 1.4 and 19.5 h, $R^2$ 0.97 and 0.55 in G2 and M, respectively. *ATF3* $t_{1/2}$ 1.9 and 29.0 h, $R^2$ 0.98 and 0.74 in G2 and M, respectively. (E) Scatterplot comparing transcript stabilization (M/G1 half-life ratio) and changes in translation efficiency (M/Async translation efficiency ratio, data from (Ly et al, 2024)). $n = 4772$, $R_s = -0.12$. (F) Probability density function plot comparing the degree of transcript stabilization (M/G2 half-life ratio) for coding transcripts and lncRNAs. Median stabilization 3.8-fold for coding transcripts and 3.1-fold for lncRNAs.

remove the m7G cap, which stabilizes the 5′ end of the mRNA (Van Dijk, 2002). The body of the transcript is then subject to digestion by 5′-to-3′ and 3′-to-5′ exonucleases (Chang et al, 2014; Anderson, 1998). Inhibition of any of these steps and their corresponding enzymes during mitosis could result in global transcriptome stabilization.

We first considered whether the most downstream steps of the mRNA degradation pathway are inactivated in mitotic cells. To assess mRNA endo- and exonuclease activity in mitotically arrested cells, we used siRNA treatment to induce cleavage of a GFP reporter construct, thereby bypassing upstream steps in the mRNA degradation pathway. We transfected GFP-expressing cells with either a non-targeting control siRNA or an siRNA designed against a site within the *GFP* transcript (Fig. 3A; Table EV1). We then arrested the cells in either G2 or M using a single thymidine block followed by treatment with RO-3306 or STLC, respectively. If RNA endo- and exonucleases were inactivated during mitotic arrest, the *GFP* mRNA or its 5′ and 3′ cleavage products would remain stable following siRNA treatment. However, if the nucleases retain activity during mitotic arrest, siRNA treatment would lead to the destabilization of the *GFP* transcript. Indeed, we found that the levels of both 5′ and 3′ fragments of the *GFP* mRNA were reduced upon siRNA treatment in mitotic cells, although not to the extent observed in G2 cells. This suggests that both 5′-to-3′ and 3′-to-5′ exonucleases remain at least partially active during mitotic arrest (Fig. 3B,C; Table EV2) and implicates upstream steps in the mitotic stabilization of mRNA.

## Deadenylation is globally perturbed during mitosis

As deadenylation is the first and rate-limiting step of the mRNA degradation pathway (Chen and Shyu, 2011; Eisen et al, 2020; Passmore and Coller, 2022), we next analyzed changes in poly(A)-tail length. To experimentally evaluate deadenylation dynamics, we conducted poly(A) tail-length profiling by sequencing (PAL-seq) (Subtelny et al, 2014) in cells arrested in mitosis or G2. To observe deadenylation dynamics, we measured poly(A)-tail lengths after up to 6 h of transcriptional inhibition (Fig. 4A). Although mitotic cells are naturally transcriptionally inactive (Fig. 1C), we treated them with the same concentration of actinomycin D as G2 cells to minimize differences between experimental conditions. In G2-arrested cells, the initial tail-length distribution was consistent with previous reports of the bulk poly(A) tail-length profile in asynchronous cells (Subtelny et al, 2014), with a peak at around 40–50 nt and a strong right skew (Fig. 4B). In the absence of new mRNA synthesis, which typically acts to replenish long-tailed species lost in ongoing deadenylation, the bulk poly(A) tail-length distribution in G2 cells shifted leftward over time, reflecting a

progressive depletion of longer-tailed species and overall tail length shortening. The shift was particularly noticeable in the 0–30 nt and >150 nt ranges (Fig. 4B), but was modest overall, presumably because many of the short-tailed species were degraded after their tails were shortened over the timecourse.

Intriguingly, the tail length distributions in mitotic cells appeared markedly different from those of G2-arrested cells (Fig. 4C). At the initial timepoint, the median poly(A)-tail length in G2 cells was 72 nt, compared to 76 nt in mitotic cells. This lack of difference at the first timepoint was expected, as mitotic cells had only just entered mitosis and transcription was not yet inhibited in G2 cells, causing tail-length distributions to reflect those observed at steady state. However, after 6 h of transcriptional inhibition, the median tail length in G2 cells decreased to 58 nt, compared to 67 nt in mitotic cells. Mitotic distributions still showed shortening of long tails in the >150 nt range after transcriptional inhibition. In contrast, the shift in the 0–30 nt range became notably less prominent, suggesting that shorter-tailed transcripts are not deadenylated as efficiently during mitotic arrest as they are in interphase. Although it is formally possible that short-tailed isoforms could be cleared more efficiently in mitotic cells, this possibility is not consistent with the mRNA stabilization that we observed during mitotic arrest. As most mRNA degradation occurs on short-tailed mRNA species, efficient clearance of such transcripts would be expected to lead to a decrease in mRNA half-life (Figs. 1B,D,E and 2B–D).

In addition to the reduction in overall poly(A) tail-length shortening, we observed distinct "bumps" in the mitotic tail length distributions, occurring with a ~30 nt periodicity as early as 0–1 h after entry into mitosis and persisting throughout a 6 h mitotic arrest (Fig. 4C). These bumps were notably absent from the spiked-in zebrafish poly(A)-tails (Fig. EV3A). A similar shape of the tail length distribution has been reported in conditions of >3-h transcription inhibition in mouse cells (Eisen et al, 2020) or following inactivation of the CCR4–NOT complex (Yi et al, 2018). As each cytoplasmic poly(A)-binding protein (PABPC) molecule occupies a ~27 nt region of the poly(A) tail (Baer and Kornberg, 1980), this distinctive patterning has been attributed to a more dense, and therefore more phased, packing of PABPC on the poly(A) tail, coupled with reduced CCR4–NOT 3′ exonuclease activity upon reaching a PABPC-protected region of the tail (Yi et al, 2018; Eisen et al, 2020). We will therefore refer to the observed periodic bumps as "phased PABPC toeprints" (noting that toeprints, in contrast to "footprints," mark only one end of a binding site (Nou and Kadner, 2000)). Consistent with prior studies (Eisen et al, 2020), subtle toeprints also began to emerge after 6 h of transcriptional inhibition in G2 cells. However, the more rapid appearance of toeprints in mitotic samples, which occurred after

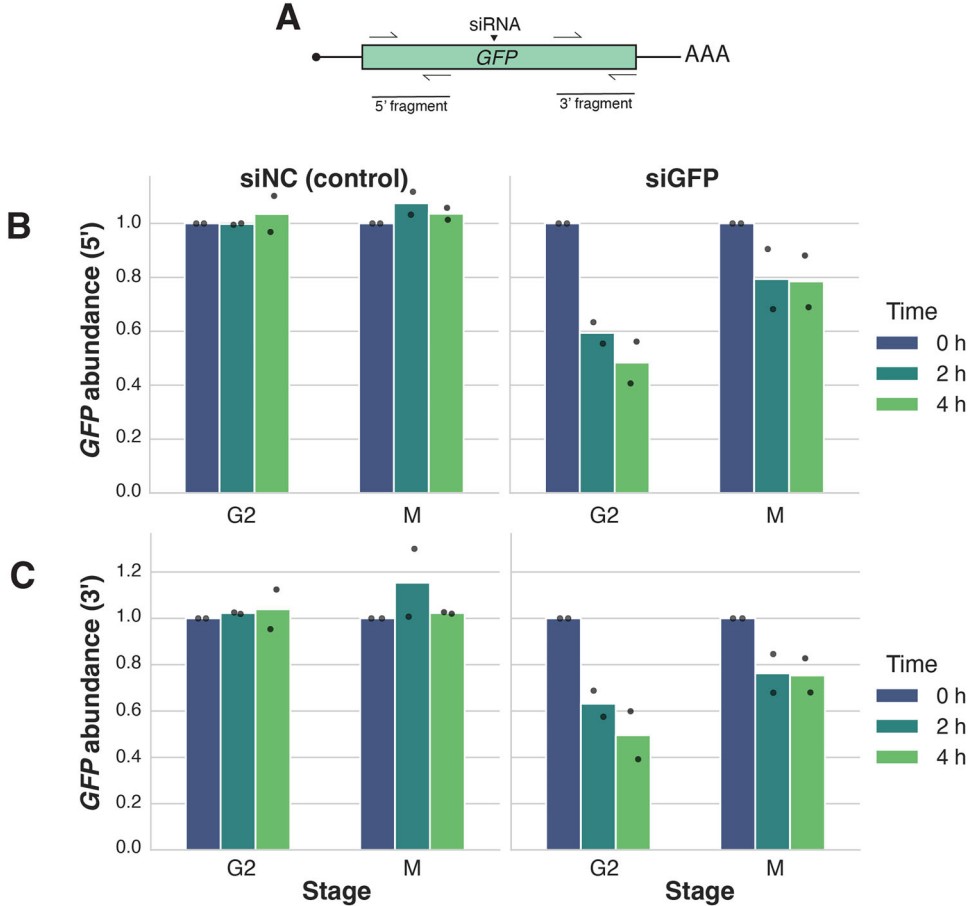

**Figure 3. siRNA-mediated mRNA degradation is active in mitosis.**

(A) Schematic diagram of the siRNA cut site and qPCR primers on the *GFP* reporter transcript. (B, C) Bar graphs comparing abundances of the (B) 5′ and the (C) 3′ fragments of the *GFP* reporter transcript upon siRNA treatment in G2- and M-arrested cells treated with actinomycin D. Mean of two biological replicates is plotted, with each point representing a biological replicate.

just 0–1 h in mitosis, suggests that a mechanism other than transcriptional inhibition feedback must be responsible for this effect in mitotically arrested cells. Importantly, the trends that we observed in the transcriptome-wide data were also apparent at the level of individual transcripts (Fig. EV3B–E). Analysis of the 25% most highly expressed transcripts and the 25% lowest expressed transcripts confirmed the reduced leftward shift in the low poly(A) tail-length range and the similar pattern of phased toeprints in each case (Fig. EV3F,G). We observed a similar behavior regardless of the degree of mitotic mRNA stabilization, with the 25% most stabilized and 25% least stabilized transcripts exhibiting the same trends (Fig. EV3H,I). Thus, the prominent periodic PABPC toeprints in mitotic samples were not driven solely by a few highly expressed genes, and did not correlate with the degree of mitotic transcript stabilization.

To assess whether deadenylation is repressed during a prolonged mitotic arrest, we also conducted PAL-seq on cells arrested in mitosis over the course of 24 h in the absence of pharmacological transcription inhibition. Tail-length distributions in these cells were consistent with those measured in transcriptionally inhibited cells (Fig. 4D). We observed the same pattern of phased PABPC toeprints as in

actinomycin D-treated cells, with well-defined bumps appearing as early as 0–1 h into mitosis and persisting for hours. Together, these observations suggest that deadenylation is perturbed in mitotically arrested cells, leading to inefficient tail-length shortening in the absence of new mRNA synthesis. The striking phasing of PABPC toeprints, which becomes apparent almost immediately as cells enter mitosis, suggests that PABPC is not efficiently removed from the poly(A) tails in this cell cycle stage (Yi et al, 2018).

## *UBE2C* escapes transcriptome-wide mitotic stabilization

The change in mRNA stability and deadenylation during mitotic arrest occurs globally and affects the vast majority of the analyzed transcripts. However, we identified *UBE2C*, a transcript encoding an E2 ubiquitin ligase, as a notable exception to the transcriptome-wide trends (Fig. 5). UBE2C functions together with the anaphase promoting complex/cyclosome (APC/C), a cell cycle-regulated E3 ubiquitin ligase that controls progression through mitosis by targeting various substrates for proteasomal degradation (Jin et al, 2008). Interestingly, in our analysis, the *UBE2C* mRNA consistently behaved contrary to the transcriptome-wide trends. In samples

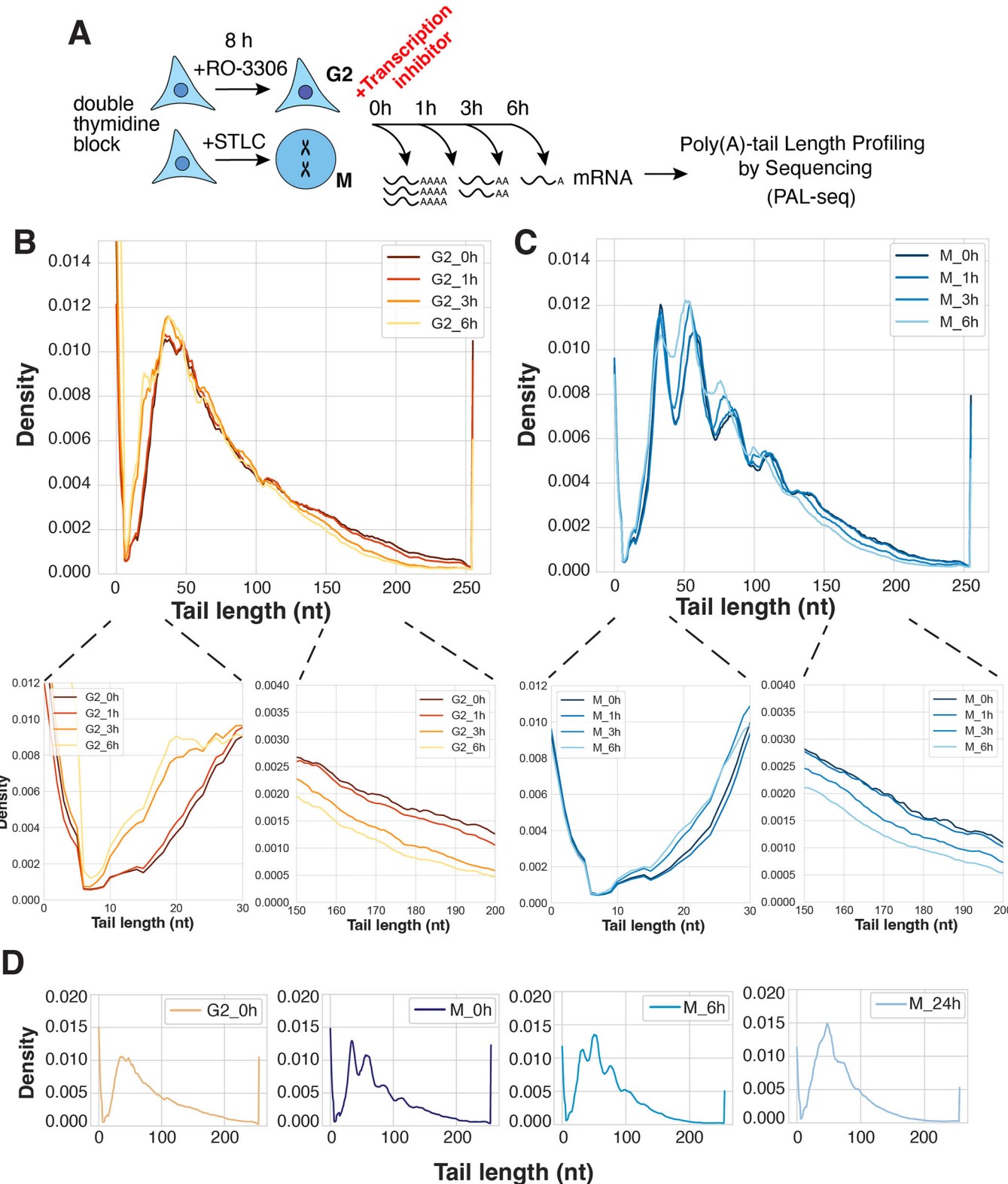

**Figure 4. Deadenylation is globally perturbed during mitotic arrest.**

(A) Schematic diagram of transcriptional inhibition timecourse experiments to measure poly(A) tail-length profiles in synchronized cells by PAL-seq. (B, C) Poly(A) tail-length distributions in (B) G2- or (C) M-arrested cells treated with actinomycin D as measured by PAL-seq. Insets show poly(A) tail-length distributions in the 0–30 nt and 150–200 nt ranges. (D) Poly(A) tail-length distributions in M-arrested cells without transcriptional inhibitors as measured by PAL-seq. A trace from G2-arrested cells prior to actinomycin D treatment (shown in B) is included for reference.

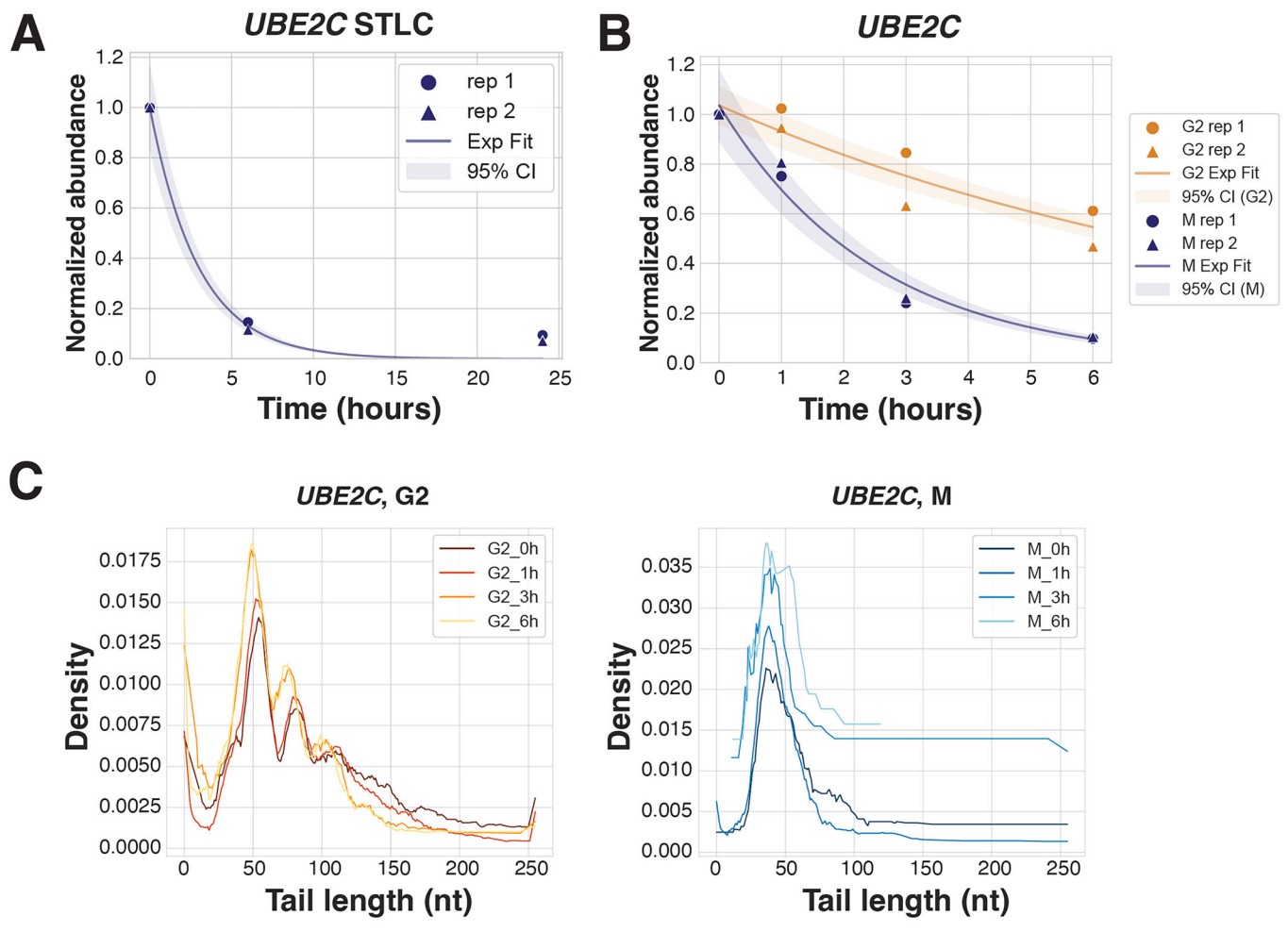

**Figure 5.** ***UBE2C* escapes transcriptome-wide mitotic stabilization.**

(A) Exponential curve fit to *UBE2C* transcript abundances over 24 h of mitotic arrest by STLC, plotted with 95% confidence intervals. $t_{1/2}$ 2.1 h, $R^2$ 0.99. (B) Exponential curve fit to *UBE2C* transcript abundances over 6 h of G2 or mitotic arrest with transcription inhibition by actinomycin D, plotted with 95% confidence intervals. $t_{1/2}$ 6.3 h and 1.8 h, $R^2$ 0.97 and 0.98 in G2 and M, respectively. (C) Poly(A) tail-length distributions in G2- or M-arrested cells treated with actinomycin D as measured by PAL-seq for *UBE2C*.

arrested in mitosis by STLC treatment for 24 h, *UBE2C* displayed rapid degradation, with a fitted $t_{1/2}$ of 2.1 h (Fig. 5A). Similarly, in transcriptionally arrested G2- and M-synchronized cells, *UBE2C* was destabilized in mitosis compared to interphase ($t_{1/2}$ of 1.8 h in M and 6.3 h in G2; Fig. 5B). Perhaps most strikingly, the distributions of poly(A)-tail lengths for this transcript were also reversed in interphase and mitosis, with prominent phasing in G2 samples and no phasing in mitotic samples (Fig. 5C). Therefore, our analysis suggests that some transcripts have evolved ways to evade transcriptome-wide stabilization during mitotic arrest.

## Possible mechanisms of mitotic mRNA stabilization

The presence of distinctive PABPC footprints could either suggest that PABPC is bound more tightly to poly(A) tails during mitotic arrest, or reflect a change in the activity of deadenylases leading to their reduced ability to displace PABPC from the mRNA. To evaluate global changes in interactions between RNA-binding proteins (RBPs) and poly(A) tails during mitotic arrest, we next measured protein association with oligo(A)-coupled beads in interphase or mitotic cell lysates. Quantitative mass spectrometry of the isolated proteins showed an enrichment of known poly(A)-interacting RBPs, such as nuclear and cytoplasmic PABP, under both conditions (Fig. EV4A,B). However, we did not detect differences in oligo(A) associations between interphase and mitotic cells, with the exception of proteins that likely represent experimental noise (Fig. 6A; Dataset EV3).

Prior studies found that the inhibition or downregulation of CNOT6 (CCR4), one of the two catalytic subunits of the CCR4–NOT complex, leads to PABPC toeprints in global poly(A) tail-length distributions (Yi et al, 2018). Thus, we hypothesized that the CCR4–NOT complex composition might be modified in mitotic cells, resulting in reduced catalytic activity of the CCR4 subunit. To test this model, we performed quantitative immunoprecipitation coupled with mass spectrometry (IP-MS) on the CCR4–NOT complex by isolating the scaffolding subunit CNOT1 using a GFP fusion. Our data did not reveal changes in subunit composition (Fig. 6B; Dataset EV4).

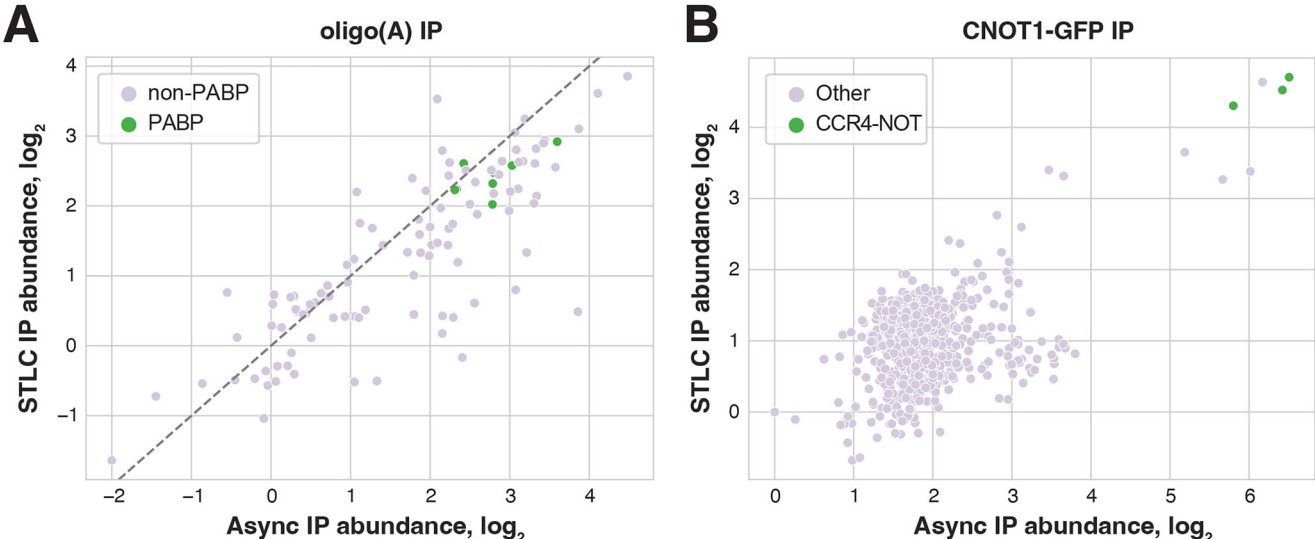

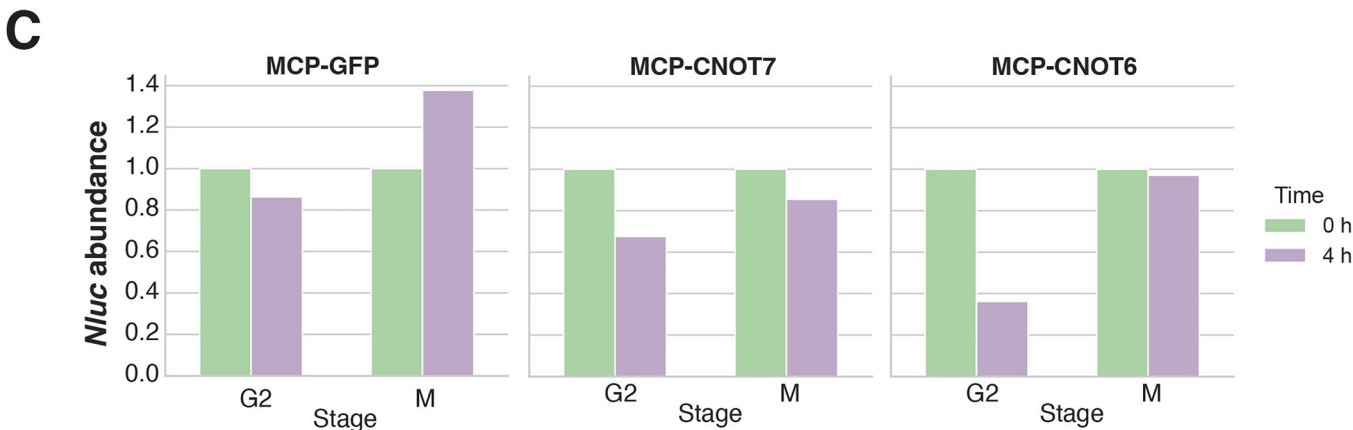

**Figure 6. Possible contributions of degradation factors to mitotic mRNA stabilization.**

(A) Scatterplot of oligo(A)-immunoprecipitated proteins in STLC-arrested or asynchronously growing cells, as measured by quantitative TMT mass spectrometry. Poly(A)-binding proteins are highlighted in green. (B) Scatterplot of CNOT1-GFP-immunoprecipitated proteins in STLC-arrested or asynchronously growing cells, as measured by quantitative TMT mass spectrometry. CCR4–NOT complex subunits are highlighted in green. Points represent averages of $n = 2$ biological replicates. (C) Bar graphs showing abundances of the *Nluc-MS2* reporter transcript in M- or G2-arrested cells upon tethering of either MCP-tagged GFP control, or MCP-GFP-tagged deadenylases CNOT7 and CNOT6 after 0 or 4 h of actinomycin D treatment.

Since our data did not show changes in CCR4–NOT complex composition, we considered whether its recruitment to transcripts could be impaired. To evaluate this possibility, we performed reporter-based assays in which we tethered the two deadenylase subunits of the complex, CNOT6 and CNOT7, to a *Nanoluciferase* reporter transcript using the MS2-MCP tethering system (Luo et al, 2020). In interphase, tethering each of these factors led to increased degradation of the reporter transcript compared to a GFP-tethered control. In contrast, neither of the tested deadenylases caused strong destabilization of the reporter mRNA in mitotically arrested cells, suggesting that recruitment of these factors to a target transcript is not sufficient to override inactivation of the degradation pathway during mitosis (Fig. 6C; Table EV2).

Notably, prior work suggested that CNOT6 knockdown produces toeprints, whereas CNOT7 knockdown does not (Yi et al, 2018). Given that the CNOT6 knockdown phenotype matched our tail-length profiles in mitotically arrested cells, we hypothesize that CNOT6 may be inhibited during mitosis to give rise to the toeprints, but CNOT7 is still active. We propose that one or both of these factors undergo post-translational modifications in mitotic cells, which could prevent their activity regardless of their binding to the target transcript.

Together, our data support a model in which deadenylation is perturbed in mitotically arrested cells, contributing to a global decrease in mRNA degradation and enabling cells to maintain their transcriptomes in the near absence of new mRNA synthesis.

Further studies are needed to more deeply evaluate the mechanistic contributions of various mRNA decay factors to this stabilization.

## PABPC1&4 contribute to mitotic transcriptome stabilization

In interphase cells, deadenylases efficiently remove PABPC from poly(A) tails to allow for tail shortening. The presence of prominent periodic toeprints prompted us to hypothesize that PABPC contributes to transcript stabilization during mitotic arrest by protecting poly(A) tails from deadenylation and consequent degradation. PABPC toeprints have been reported in conditions in which CCR4–NOT activity is impaired. According to the current model of CCR4–NOT-mediated deadenylation, two catalytic complex subunits (CNOT6 and CNOT7) have differing activities: the former is capable of displacing PABPC and degrading the underlying poly(A) sequence, and the latter only degrades free poly(A) (Yi et al, 2018). As CNOT6 inactivation leads to the emergence of PABPC toeprints due to CNOT7 still degrading free poly(A) but stopping at PABPC-bound poly(A), we wondered whether enabling further CNOT7-mediated deadenylation by artificially depleting PABPC would restore mRNA degradation in mitotic cells. Additionally, we reasoned that mRNA degradation would only be restored in PABP-depleted mitotic cells if mRNA degradation machinery that acts downstream of deadenylation, such as decapping enzymes, retained their activity in mitosis. In contrast, if these enzymes were completely inactivated during mitotic arrest, PABPC1&4 depletion would be expected to have no effect on mitotic mRNA stability, as those factors would no longer be able to clear deadenylated transcripts.

To assess this hypothesis, we used a PABPC1&4-Auxin-Inducible Degron (AID) system (Xiang and Bartel, 2021; Fig. EV5A) and evaluated the effects of rapid PABPC1&4 co-depletion on mRNA stability in mitotic cells. To selectively eliminate PABPC1&4 in mitotically arrested cells without affecting prior interphase behaviors, we synchronized cells using STLC treatment, after which IAA was added to induce rapid OsTIR1-mediated degradation of PABPC1&4 (Fig. 7A,B). In control cells expressing PABPC1&4, mRNA levels did not change substantially after 4 h of mitotic arrest, consistent with our original finding (Fig. 7C,D). Strikingly, in PABPC1&4-depleted cells, the levels of most mRNAs decreased within 4 h of IAA addition, suggesting that the mRNA degradation pathway is active in mitotic cells in the absence of PABP. This result suggests that PABP depletion can restore deadenylation activity during mitotic arrest, and that PABP-dependent inhibition of deadenylation contributes to the global transcriptome stabilization observed in mitotically arrested cells.

## mRNA stabilization is necessary for the maintenance of mitotic arrest

Cells arrested in mitosis using STLC treatment or other mitotic disruptions can remain viable for tens of hours despite the near absence of transcription. In mitotically arrested cells, ongoing translation is necessary for the maintenance of mitotic arrest (Malureanu et al, 2010; Mena et al, 2010; Sloss et al, 2016; Tsang and Cheeseman, 2023; Ly et al, 2024). Therefore, we hypothesized that mitotic transcriptome stabilization would also be required for cells to continue translation of key mitotic factors and thus persist

in the absence of new transcription during mitotic arrest. To assess the physiological importance of mitotic transcriptome stabilization, we tracked the fate of STLC-treated HCT116 cells in the presence or absence of PABPC1&4 using the AID system (Fig. 7E).

Cells challenged with STLC face two mutually exclusive fates: they either die during mitosis or exit mitosis without chromosome segregation, entering G1 through a process known as "mitotic slippage" (Skoufias et al, 2006). In contrast to control cells that remained arrested in mitosis for hours, PABPC1&4 depletion resulted in >2-fold accelerated mitotic slippage (Fig. 7F,G, median times of 820 min for control vs 395 min for PABP-depleted cells). Notably, PABPC1&4 depletion does not affect translation efficiency (Xiang and Bartel, 2021). Therefore, the effect of PABPC1&4 depletion on mitotic slippage is likely to be mediated primarily by its effects on deadenylation and mRNA stability, as established here and in the literature. Thus, in the absence of transcriptome stabilization, cells are unable to maintain a mitotic arrest to fix cell division errors and instead "slip out" into a tetraploid G1 state without dividing.

Together, these results suggest that repression of deadenylation in mitotic cells contributes to globally preserving mRNA levels during an extended mitotic arrest. Crucially, if cells cannot stabilize mRNA in mitosis, their ability to maintain a prolonged arrest is significantly impaired, suggesting that this mechanism is critical for proper mitotic physiology.

## Discussion

Our study uncovers a previously unappreciated phenomenon in which mammalian cells maintain a state of mitotic arrest for tens of hours without global transcriptome changes. We demonstrate that mRNA levels are maintained over a 24-h mitotic arrest period in the near absence of transcription. The transcriptome is preserved due to a global increase in mRNA stability in mitotic cells. This mRNA stabilization is mediated, at least in part, by the attenuation of deadenylation, as evident from slowed shortening of poly(A)-tail lengths, characteristic phasing of PABPC-protected toeprints, and the dependence of this stabilization on the presence of PABPC1&4.

Prior reports suggested that there is a transcriptome buffering feedback, with cells in which transcription is inhibited for 24 h adapting to this change by reducing mRNA decay (Slobodin et al, 2020). Although such a feedback mechanism could be partially responsible for the stabilization in mitotically arrested cells observed in this study, this is inconsistent with the rapid changes we observed in both mRNA half-lives and poly(A) tail-length profiles upon entry into mitosis. For instance, in prior studies of transcription-degradation feedback, the effect of transcription perturbation on poly(A)-tail profiles did not occur until >3 h after drug treatment in NIH 3T3 cells (Eisen et al, 2020). In contrast, in mitotically arrested cells, the characteristic phased PABPC toeprints become clear as early as 0–1 h after entry into mitosis (Fig. 4). Additionally, if transcriptional inhibition was solely responsible for mRNA stabilization in mitotic cells, our experiments comparing mRNA half-lives in cells with pharmacologically inhibited transcription would have shown similar results in cells arrested in G2- or M-phase. Therefore, a mechanism other than transcription-degradation feedback must contribute to prompt stabilization of the transcriptome upon entry into mitosis.

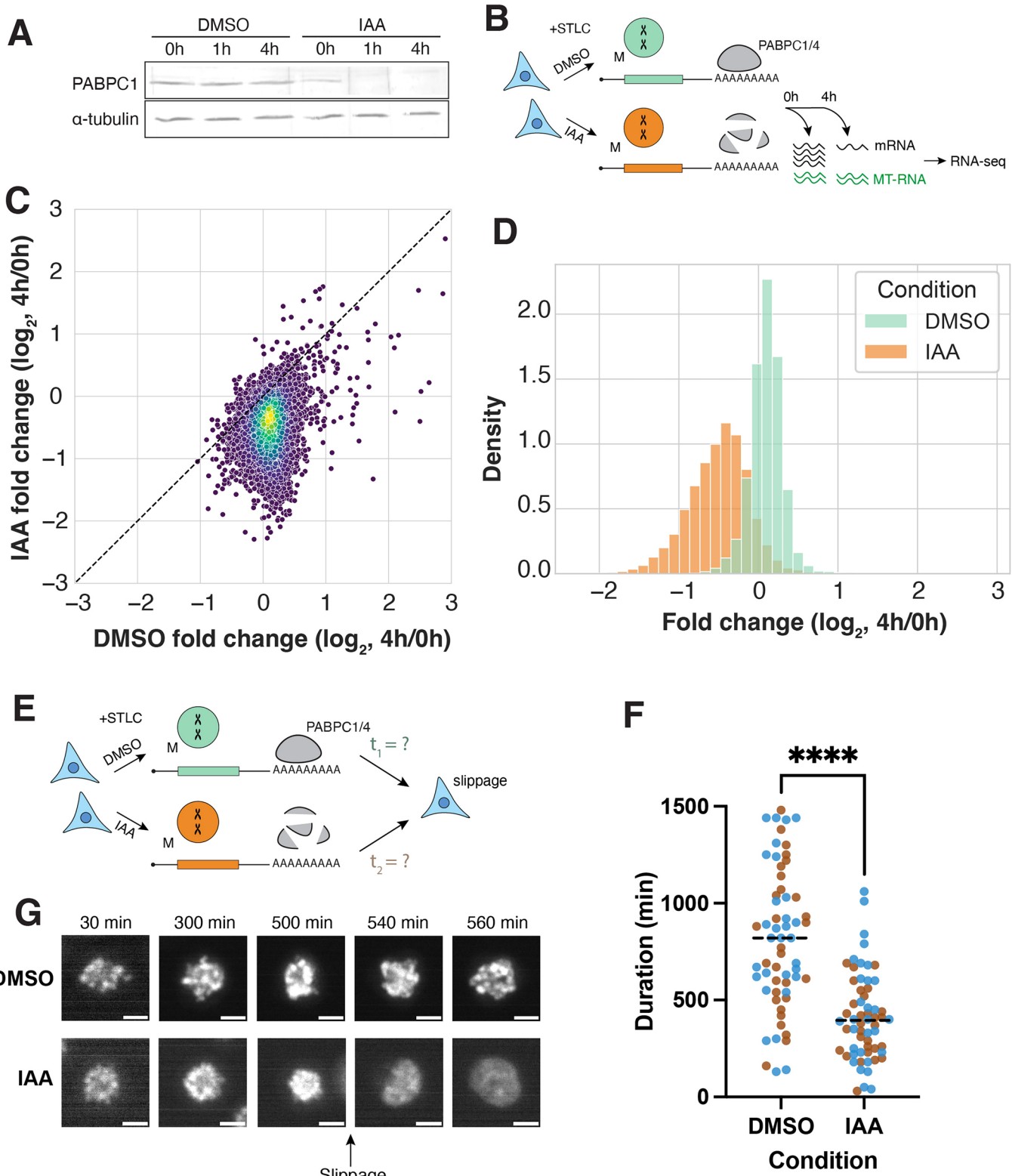

Prior work found that the median poly(A)-tail length remains largely unchanged in cells arrested in interphase or mitosis (Park et al, 2016). Given the lack of transcription in mitotically arrested cells, if deadenylation occurred at similar rates in these two conditions, mitotic poly(A)-tail lengths would be expected to be

shorter than those in interphase after several hours of arrest. In this case, deadenylation would lead to tail shortening without the concurrent replenishing of long-tailed transcript species that occurs in transcriptionally active interphase cells. Therefore, the lack of a global difference in median poly(A)-tail lengths reported by Park

**Figure 7.  PABPC1&4 are required for mitotic transcriptome stabilization and maintenance of mitotic arrest.**

(A) Western blot showing PABPC1 depletion in STLC-arrested cells following either DMSO treatment, or IAA-induced degradation of PABPC1&4. alpha tubulin shown as a loading control. (B) Schematic diagram of sequencing experiments with PABPC1&4 depletion in M-arrested cells. (C) Scatterplot comparing mRNA abundance fold-changes in M-arrested HCT116 cells within 4 h of either DMSO control treatment or IAA-induced PABPC1&4 depletion. Each point represents the mean of two replicates, $n = 11{,}078$. Median fold-changes: 1.07 and 0.72 for control and PABPC1&4 depletion, respectively. (D) Probability density function plot comparing expression fold-change within 4 h of either DMSO control treatment or IAA-induced PABPC1&4 degradation. (E) Schematic diagram of live imaging experiments with PABPC1&4 depletion to measure mitotic slippage in STLC-arrested cells. (F) Dot plot comparing time from control treatment or PABPC1&4 degradation until mitotic slippage in STLC-treated HCT116 cells. $n = 30$ cells per condition per replicate, points colored by replicate. Median durations: 820 min for DMSO control cells and 395 min for PABPC1&4-depleted cells. Student's $t$-test $p = 2.65318\text{E-}11$. (G) Representative images of SiR-DNA-stained DNA in STLC-arrested HCT116 cells with (DMSO) or without (IAA) PABPC1&4. Brightness not scaled identically to highlight differences between condensed and de-condensed chromosomes. Source data are available online for this figure.

et al is consistent with our findings of impaired deadenylation in mitotic cells.

We hypothesize that inactivation of CNOT6 may contribute to mRNA stabilization during mitotic arrest. Although altered polyadenylation could also contribute to differences in poly(A)-tail profiles, recent work showed that mitotic phosphorylation of PABPN1 prevents hyperadenylation (Gordon et al, 2024). Therefore, decreased deadenylation by CCR4–NOT is the most likely explanation for the striking poly(A) tail-length distributions in mitotically arrested cells. Consistent with our model, prior work demonstrated that inactivation of CNOT6 results in phased PABPC toeprints similar to the ones observed in our study (Yi et al, 2018). Additionally, a recent study suggested that the CCR4–NOT complex might not associate with RNA as efficiently in mitosis as it does in interphase (Fig. EV5C; (Rajagopal et al, 2025).

Importantly, targeted depletion of PABPC1 and 4 during mitotic arrest leads to a strong reduction in mRNA stability and also prevents cells from maintaining a mitotic arrest. Mechanistically, the effect of PABPC1&4 depletion on mitotic arrest maintenance is likely to involve the combined effects of destabilizing hundreds of mRNAs, with specific contributions from each transcript. For example, we propose that one relevant target of destabilization in the absence of PABPC is the *CCNB1* (Cyclin B1) transcript. Continued CCNB1 translation during mitosis is required for sustaining the mitotic state through cyclin-dependent kinase activity, with CCNB1 degradation serving as a key step in mitotic exit (Mena et al, 2010; Malureanu et al, 2010). We found that the *CCNB1* transcript stability is increased 6-fold in mitotic cells compared to G2. Additionally, *CCNB1* is 36% less stable in PABPC-depleted cells compared to control cells (Fig. EV5D). Although these observations do not demonstrate a causative role of *CCNB1* mRNA degradation in the early slippage of PABPC-depleted mitotic cells or preclude contributions from other mRNAs, they are consistent with the expectations based on previously established effects of CCNB1 depletion on the maintenance of the mitotic state (Mena et al, 2010).

Together, our findings reveal a novel gene-regulatory framework in mitotic cells that is responsible for maintaining mRNA levels required for cells to endure periods of transcriptional silencing. Importantly, our results could have therapeutic implications, as anti-mitotic drugs such as paclitaxel are widely used as frontline cancer therapeutics (van Vuuren et al, 2015). Although such drugs are effective at specifically targeting dividing cells, they are also known to cause a range of adverse side effects, highlighting a need for improvements in their efficacy (Cella et al, 2003; Awosika et al, 2025). Our study nominates PABP and the mRNA degradation machinery as potential targets to enhance the efficacy of such drugs.

# Methods

### Reagents and tools table

| Reagent/resource | Reference or source | Identifier or catalog number |
|---|---|---|
| **Experimental models** | | |
| HeLa cells (human) | Cheeseman lab | N/A |
| HEK293T (human) | Cheeseman lab | N/A |
| mEGFP HeLa (human) | This study | cJL217 |
| HeLa CAG::Nluc-5xMS2, EIF1A::Fluc (human) | This study | cKK10 |
| MCP-GFP HeLa CAG::Nluc-5xMS2, EIF1A::Fluc (human) | This study | cKK23 |
| MCP-GFP-CNOT6 HeLa CAG::Nluc-5xMS2, EIF1A::Fluc (human) | This study | cKK16 |
| MCP-GFP-CNOT7 HeLa CAG::Nluc-5xMS2, EIF1A::Fluc (human) | This study | cKK12 |
| GFP-CNOT1 | This study | cKK6, cKK7 clones |
| PABPC1&4-AID HCT116 (human) | This study | sC278-287-C8 |
| **Recombinant DNA** | | |
| mEGFP in lentiviral vector | This study | pJL347 |
| CNOT1-GFP in lentiviral vector | This study | pKK17 |
| Nluc-MS2 Fluc in pBMWb112, a modified PiggyBac donor plasmid (System Biosciences), gifted by the Bartel lab | This study | pKK36 |
| MCP-GFP in lentiviral vector | This study | pKK19 |
| MCP-GFP-CNOT6 in lentiviral vector | This study | pKK51 |
| MCP-GFP-CNOT7 in lentiviral vector | This study | pKK35 |
| PABPC4 sgRNA | This study | C275 |
| PABPC4-AID | This study | C287 |
| **Antibodies** | | |
| anti-PABP1 | Cell Signaling Technology | #4992 |
| anti-PABP4 | Thermo Fisher Scientific | NB100-74594 |
| anti-beta actin | Cell Signaling Technology | D6A8 |
| anti-vinculin | Proteintech | 66305 |
| anti-alpha tubulin | Sigma-Aldrich | T9026 |

| Reagent/resource | Reference or source | Identifier or catalog number |
|---|---|---|
| IRDye 680RD Goat anti-Mouse | LI-COR | 92668070 |
| IRDye 800CW Goat anti-Rabbit | LI-COR | 92632211 |
| anti-phospho-histone 3 serine 10 | Abcam | 5176 |
| anti-Rabbit Cy5 | Jackson ImmunoResearch Laboratories | 111-175-144 |
| **Oligonucleotides and other sequence-based reagents** | | |
| siNC (cocktail) | Dharmacon | D-001206-13-20 |
| siGFP | Thermo Fisher Scientific | CHEIE-000017 |
| Nluc Forward | This study | Table EV2 |
| Nluc Reverse | This study | Table EV2 |
| GAPDH Forward | This study | Table EV2 |
| GAPDH Reverse | This study | Table EV2 |
| GFP 3' Forward | This study | Table EV2 |
| GFP 3' Reverse | This study | Table EV2 |
| GFP 5' Forward | This study | Table EV2 |
| GFP 5' Reverse | This study | Table EV2 |
| Biotin-(rA)$_{41}$ | This study | N/A |
| **Chemicals, enzymes and other reagents** | | |
| Doxycycline | Sigma-Aldrich | Cat#D9891 |
| S-trityl-L-cysteine (STLC) | Sigma-Aldrich | Cat#164739 |
| Thymidine | Sigma-Aldrich | T9250 |
| RO-3306 | Millipore-Sigma | SML0569-5MG |
| X-tremeGENE 9 | Roche | C762Q75 |
| Lipofectamine 2000 | Invitrogen | CAT11668 |
| Puromycin | Gibco | A1113802 |
| Actinomycin D | Life Technologies Corporation | A7592 |
| THZ1 | Thermo Fisher Scientific | 5323720001 |
| 5-EU | Vector Labs | CCT-1261 |
| Phasemaker™ tubes | Invitrogen | A33248 |
| Lipofectamine RNAiMAX Transfection Reagent | Invitrogen | 13778075 |
| cOmplete protease inhibitor cocktail | Roche | cat# 1 836 170 |
| SUPERaseIn RNase Inhibitor | Life Technologies Corporation | AM2696 |
| SYBR Green PCR Master Mix | Thermo Fisher Scientific | A25742 |
| TRIzol Reagent (Tri Reagent solution) | Life Technologies Corporation | AM9738 |
| Auxinole | Aobious | AOB8812 |
| SiR-DNA | VWR International LLC | MSPP-CYSC007 |

| Reagent/resource | Reference or source | Identifier or catalog number |
|---|---|---|
| **Software** | | |
| FlowJo v10.9.0 | https://www.flowjo.com/ | N/A |
| GraphPad Prism | https://www.graphpad.com/ | N/A |
| Jupyter notebook | https://jupyter.org/ | N/A |
| FIJI/ImageJ | https://imagej.net/software/fiji/ | N/A |
| Cutadapt v4.8 | https://cutadapt.readthedocs.io/ | N/A |
| STAR v2.7.1a | https://github.com/alexdobin/STAR | N/A |
| htseq v1.99.2 | https://htseq.readthedocs.io | N/A |
| Proteome Discoverer 2.4 | Thermo Fisher Scientific | N/A |
| Image Studio | LICORbio | N/A |
| **Other** | | |
| TMT Pro 16-plex Isobaric Labeling Reagent Set | Life Technologies Corporation | A44521 |
| The Pierce High pH Reversed-Phase Peptide Fractionation Kit | Thermo Fisher Scientific | 84868 |
| LLC S-TRAP micro kit | ProtiFi | NC1828286 |
| Click-iT™ RNA Alexa Fluor™ 594 Imaging Kit | Thermo Fisher Scientific | C10330 |
| KAPA RNA HyperPrep Kit with RiboErase (HMR) | Roche | KK8560 |
| Watchmaker QiaSeq FastSelect Ribodepletion kit | Watchmaker Genomics | 7BK0002 |
| Qubit™ RNA Broad Range Assay Kit | Invitrogen | Q10210 |
| Maxima First Strand cDNA Synthesis Kit for RT-qPCR | Thermo Fisher Scientific | K1671 |
| Illumina NovaSeq SP | Illumina | |
| Element AVITI | Element Biosciences | |
| Eclipse Orbitrap mass spectrometer | Thermo Fisher Scientific | |
| Odyssey Clx | LI-COR | |
| Nikon Eclipse microscope equipped with a sCMOS camera (ORCA-Fusion BT, Hamamatsu) | Nikon | |

## Tissue culture

HeLa cells were cultured in Dulbecco's modified Eagle medium (DMEM) supplemented with 10% heat-inactivated fetal bovine serum, 2 mM L-glutamine and 100 U ml$^{-1}$ penicillin-streptomycin at 37 °C with 5% $CO_2$. HCT116 cells were cultured in McCoy's 5 A (modified) medium supplemented with 10% tetracycline-free heat-

inactivated fetal bovine serum, 2 mM L-glutamine and 100 U ml$^{-1}$ penicillin-streptomycin at 37 °C with 5% $CO_2$. Cell lines were regularly tested for mycoplasma contamination.

## Cell line generation

For siRNA transfection experiments, a HeLa cell line with ectopically expressed mGFP was used. To generate mEGFP lentivirus, HEK293T cells at ~90% confluency in a six-well plate were transfected with 1.2 μg mEGFP lentiviral transfer plasmid, 1 μg psPAX2 packaging plasmid, and 0.4 μg vsFULL envelope plasmid using X-tremeGENE 9 (Roche C762Q75). Sixteen hours after transfection, the medium was replaced. Viral supernatant was collected 24 h later and stored at −80 °C. HeLa cells at ~90% confluency in a six-well plate were incubated with 100 μL mEGFP lentivirus and 10 μg/mL polybrene. After 16 h, the medium was replaced. Two days post-infection, GFP-positive cells were bulk-sorted using a BD FACS Aria III cell sorter.

For RBP tethering assays, first, a parental CAG::Nluc-5xMS2, EIF1A::Fluc HeLa cell line was created by piggyBac transfection. HeLa cells were grown to 60% confluency and transfected with 1 μg Nluc-Fluc donor plasmid and 400 ng piggyBac transposase plasmid using Lipofectamine 2000 (Invitrogen CAT11668) transfection reagent. Sixteen hours after transfection, the medium was replaced. Transfected cells were selected by 0.35 μg/mL puromycin for 7 days. MCP-GFP-tagged RBP constructs were then inserted using lentiviral transfection, as described above. GFP-positive cells were bulk-sorted using a BD FACS Aria III cell sorter.

For the CNOT1-GFP pulldown assay, the CNOT1-GFP lentivirus was generated, and HeLa cells were infected as described above. GFP-positive cells were single-cell sorted using a BD FACS Aria III cell sorter. Two clones were selected for the pulldown experiments.

## Cell cycle synchronization

For experiments done with a double thymidine block, HeLa cells were grown to 20% confluency and treated with 2 mM thymidine for 18 h. Cells were released from thymidine arrest by washing twice with warm DMEM and cultured for 9 h without drugs before a second treatment with 2 mM thymidine for another 14 h. Cells were released from thymidine arrest by washing twice with warm DMEM and cultured in 8 μM STLC or 6 μM RO-3306 for 8 h until ~50% of STLC-treated cells were rounded. Mitotic cells were isolated by shaking off. Actinomycin D was added to cells at a concentration of 5 μg/mL. THZ1 was added to cells at 1 μM. For G1 synchronization, cells were synchronized using the double thymidine block procedure, followed by release into 8 μM STLC for 8 h. Mitotic cells were shaken off and replated in DMEM and allowed to progress to G1 and adhere to the plate for 1 h before transcriptional inhibitor treatment. For experiments done with a single thymidine block, HeLa cells were grown to 40% confluency and treated with 2 mM thymidine for 24 h before being released into 8 μM STLC as described above. For qPCR experiments, actinomycin D was added, and the first timepoints were collected after 16 h of STLC treatment. For experiments with the HCT116 PABP AID cell line, cells were grown to 20% confluency and treated with 8 μM STLC for 16 h.

## Cell cycle quantification with FACS

Mitotic cells were collected by shaking off. Interphase cells were detached with 5 mM EDTA in PBS. All cells were then washed once with PBS and spun down at 500×$g$ for 5 min. Cells were then resuspended in 500 uL PBS and vortexed while 4.5 mL of −20C 100% ethanol was added dropwise. After a 30-min incubation on ice or overnight storage at 4 °C fixed cells were spun down at 1000×$g$ for 5 min and resuspended in 500 uL ethanol with 9.5 mL cold PBS with 3% BSA to facilitate pelleting and prevent clumping. Cells were then washed once with 1 mL PBS + 3% BSA + 0.1% Triton X-100 and incubated on ice in 1 mL antibody dilution buffer (20 mM Tris-HCl, 150 mM NaCl, 0.1% Triton X-100, 3% bovine serum albumin, 0.1% NaN$_3$, pH 7.5) for 30 min. Cells were then incubated in 300 uL antibody dilution buffer with 1:1000 phospho-histone 3 serine 10 antibody (Abcam 5176; Rabbit) with slow rotation overnight at 4 °C. After a 1 mL wash in PBS + 3% BSA + 0.1% Triton X-100, cells were incubated for 1 h on ice with 300 μL antibody dilution buffer with 1:300 anti-Rabbit Cy5 antibody (Jackson ImmunoResearch Laboratories; Goat). Cells were then washed once with PBS + 3% BSA + 0.1% Triton X-100, resuspended in 500 uL PBS + 20 ug/mL Hoechst and incubated on ice for 30 min before being strained into flow cytometry tubes. Analysis of cell cycle stage was done using the BD FACSymphony A1 Cell Analyzer (BD Biosciences) and quantified using FlowJo v10.9.0. We first gated cells using forward scatter and side scatter area, followed by identifying cells with 4n DNA content and classifying them as either G2 (pH3S10-negative) or M (pH3S10-positive).

## Measurement of global transcription rate with FACS

HeLa cells were synchronized by double thymidine block followed by STLC treatment as described above. 0, 5, or 23 h after the cells entered mitotic arrest, 1 mM 5EU was added to the cells for 1 h. Asynchronous control cells received the same treatment, with the addition of 5 ug/mL of actinomycin D to one of the wells. After 1 h of 5EU labeling, mitotic cells were harvested by shake-off and asynchronous cells were dissociated from wells with 5 mM PBS-EDTA. Cells were washed with 1 mL cold PBS and fixed with 1 mL of 3.7% formaldehyde in PBS for 15 min at room temperature, followed by a wash in 1 mL cold 2.5% PBS-BSA. Cells were permeabilized in 1 mL 0.5% Triton X-100 in PBS for 15 min at room temperature. Click chemistry was then performed using the Click-iT™ RNA Alexa Fluor™ 594 Imaging Kit (Thermo Fisher Scientific) according to the manufacturer's instructions, substituting coverslip washes with 4 min 500×$g$ spins at 4 °C. After click chemistry, cells were resuspended in 500 uL antibody dilution buffer (20 mM Tris-HCl, 150 mM NaCl, 0.1% Triton X-100, 3% bovine serum albumin, 0.1% NaN$_3$, and pH 7.5) and strained into flow cytometry tubes. Analysis of 5EU incorporation was done using the BD FACSymphony A1 Cell Analyzer (BD Biosciences) and quantified using FlowJo v10.9.0. We first gated cells using forward scatter and side scatter area, followed by plotting histograms of 5EU levels to visualize active transcription during the hour prior to cell harvesting.

## RNA isolation and sequencing

Interphase cells were detached from plates using PBS with 5 mM EDTA. Mitotic cells were collected by shake-off. All cells were washed once with PBS, and resuspended in TRIzol Reagent. All

RNA isolations were performed with Phasemaker™ tubes according to the manufacturer's protocol (Invitrogen). Total RNA was resuspended in water with 4 pg spike-in RNA (equal mixture of in vitro transcribed Fluc and Nluc) per 2 μg total RNA for experiments where spike-in RNA was used for normalization. This normalization strategy relies on the assumption that total cellular RNA levels would not drastically change over the course of the experiments. We think this is a reasonable assumption, as rRNA, which is known to be highly stable ($t_{1/2}$ on the order of multiple days in mammalian cells), comprises >95% of total RNA (Defoiche et al, 2009; Nikolov et al, 1983; Hirsch and Hiatt, 1966). Additionally, in our experiments, the cells were cell cycle-arrested, and therefore, the rRNA was not undergoing dilution. Concentration measurements were carried out using Qubit™ RNA Broad Range Assay Kit (Invitrogen).

For HeLa RNA sequencing experiments, rRNA was depleted, and libraries were prepared using the KAPA RNA HyperPrep Kit with RiboErase (HMR) (Roche KK8560) according to the manufacturer's instructions. Sequencing was performed on an Illumina NovaSeq SP, in a paired-end 50 × 50 mode.

For HCT116 RNA sequencing, total RNA was isolated as described above. rRNA was depleted and libraries prepared using the Watchmaker QiaSeq FastSelect Ribodepletion kit according to the manufacturer's instructions. Sequencing was performed on Element AVITI, in a paired-end 75 × 75 mode.

For all sequencing experiments reported, poly(A) reads were trimmed using cutadapt (v4.8; (Martin, 2011)) with the parameters '--minimum-length 1 -a A{25} -A A{25}'. Reads were then mapped to the human genome (Gencode release 25, GRCh38.p7, downloaded from the GENCODE website) using STAR (v2.7.1a; (Dobin et al, 2013) with the parameters "--runThreadN 24 --runMode alignReads --outFilterMultimapNmax 1 --outFilterType BySJout --outSAMattributes All --outSAMtype BAM SortedByCoordinate". Exon mapping reads were quantified using htseq-count (v1.99.2; Anders et al, 2015) with the parameters "-f bam -t exon -r pos -s reverse". A read cutoff of ≥25 reads in each sample was applied for each gene.

## Half-life calculations

mRNA read counts were normalized to spike-in or mitochondrial RNA read counts in pharmacologically transcriptionally inhibited and uninhibited cells, respectively. Normalized abundance values were averaged between two biological replicates and fit to the exponential decay function:

$$abundance(t) = abundance_{SteadyState} * e^{-k_d * t}$$

$$t_{1/2} = \frac{\ln(2)}{k_d}$$

Transcripts with the fit $R^2 < 0.8$ in interphase cells were excluded from subsequent analysis (Fig. EV2B). Calculated half-life values were arbitrarily capped at 200 h for half-lives ≥200 h, or 0.25 h for half-lives ≤0.25 h, for ease of visualization due to their lack of biological interpretability. For the THZ1 experiment (Fig. EV2C), due to technical considerations, normalization was done using read counts for the top 70 most stable transcripts instead of spike-in read counts, and a fit cutoff of G2 $R^2 < 0.4$ was used.

## siRNA treatment

mGFP-expressing cells were synchronized using a single thymidine block as described above. About 50 nM siRNA with RNAiMax transfection reagent was added to cells in OPTIMEM medium at the same time as STLC or RO-3306. After 16 h, the first timepoints were collected. For the remaining timepoint, the STLC-treated cells were shaken off and replated in DMEM with STLC and actinomycin D for 4 h. Fresh DMEM with RO-3306 and actinomycin D was added to the RO-3306-treated cells for 4 h. Cells were collected, and RNA was isolated as described above.

## PAL-seq

Sequencing of endogenous mRNA poly(A)-tail lengths in interphase and mitosis was performed with PAL-seq v4 as described (Xiang and Bartel, 2021). The poly(A)-tail length was determined using a Hidden Markov Model trained on randomly picked 1% of the filtered read clusters (but no more than 50,000 and no less than 5000) for each library as described (Xiang et al, 2024).

## Oligo(A) immunoprecipitation

Biotin-(dT)$_{18}$ was dissolved in Wash/Binding buffer (0.5 M NaCl, 20 mM Tris-HCl (pH 7.5), 1 mM EDTA) to a final concentration of 8 pmol/μl. Biotin-(rA)$_{41}$ was diluted to 10 μM in Wash/Binding buffer. About 50 μg of Pierce beads per IP sample were washed with Wash/Binding buffer and vortexed to resuspend. Half of the beads were incubated with 60 μL 10 μM biotin-(rA)$_{41}$ solution, and the other half were incubated with an equal volume of Wash/Binding buffer alone for 5 min at room temperature with occasional agitation by hand. Beads were then washed twice with 100 μL Wash/Binding buffer.

Asynchronously growing cells were detached from a 1 × 15 cm plate with 5 mM EDTA in PBS. A 15 cm plate of mitotic cells that were arrested using a single thymidine block and STLC treatment was harvested by shake-off as described above. Cells were then spun down at 1000 rpm for 5 min, followed by a wash in 20 mL cold PBS. The cell pellets were then washed in 20 mL cold 50 mM HEPES, pH 7.4, 1 mM EGTA, pH 8.0, 300 mM KCl, 1 mM MgCl$_2$, 10% glycerol, resuspended in ~250 μL of the same buffer, flash-frozen and stored at −80 °C. Equal volume of 1.5x lysis buffer (75 mM HEPES, pH 7.4, 1.5 mM EGTA, pH 8.0, 150 mM KCl, 1.5 mM MgCl$_2$, 15% glycerol, and 0.075% NP-40), supplemented with 1× cOmplete protease inhibitor cocktail (Roche cat# 1 836 170), 20 mM beta-glycerophosphate, 0.4 mM sodium orthovanadate, 5 mM NaF, 200 mM PMSF, and 1:1000 SUPERaseIn RNase Inhibitor was added to the frozen lysates, which were then quickly thawed in 37 °C water bath until mostly melted. Samples were then sonicated with a Branson Digital Sonifier tip sonicator at 20% amplitude for 15 s in a 5 s ON 50 s OFF regime. Lysates were then spun at 1300×$g$ for 10 min at 4 °C, after which 10% of the total volume was set aside as input and lyophilized, and 90% of the volume was combined with oligo(A)-coupled beads for immunoprecipitation.

About 100 μg beads were resuspended in 225 μl of the lysate and rotated for 2 h at 4 °C. Beads were washed 4 × 5 min with 500 μL high salt lysis buffer (300 mM KCl, 0.05% NP-40, 1:1000 DTT, 1:1000 LPC, 20 mM beta-glycerophosphate, 0.4 mM sodium

orthovanadate, 5 mM NaF, 200 mM PMSF, and 1:1000 SUPER-aseIn RNase Inhibitor).

During analysis, protein interactors were considered oligo(A)-specific if their PSM values in the IP condition were at least twofold higher than in the negative control condition.

## CNOT1-GFP immunoprecipitation

Anti-GFP nanobodies were coupled to magnetic beads as described by Marescal and Cheeseman, 2025. $10 \times 15$ cm plates of CNOT1-GFP cells and mGFP control cells were grown to ~90% confluency (asynchronous condition), and $15 \times 15$ cm plates of cells were grown and synchronized in mitosis using a single thymidine block and STLC treatment, as described above. Asynchronous cells were dissociated from the plates using 5 mM EDTA in PBS. Mitotic cells were shaken off as described above. Cells were then spun down at 1000 rpm for 5 min, followed by a wash in 20 mL cold PBS. The cell pellets were then washed in 20 mL cold 50 mM HEPES, pH 7.4, 1 mM EGTA, pH 8.0, 300 mM KCl, 1 mM MgCl$_2$, 10% glycerol, resuspended in ~800 µL of the same buffer, flash-frozen and stored at −80 °C. Equal volume of 1.5x lysis buffer (75 mM HEPES, pH 7.4, 1.5 mM EGTA, pH 8.0, 150 mM KCl, 1.5 mM MgCl$_2$, 15% glycerol, and 0.075% NP-40), supplemented with 1× cOmplete protease inhibitor cocktail (Roche cat# 1 836 170), 20 mM beta-glycerophosphate, 0.4 mM sodium orthovanadate, 5 mM NaF, and 200 mM PMSF, was added to the frozen lysates, which were then quickly thawed in 37 °C water bath until mostly melted. Samples were then sonicated with a Branson Digital Sonifier tip sonicator at 20% amplitude for 30 s in a 10 s ON 50 s OFF regime. Lysates were then spun at 21,000×$g$ for 30 min at 4 °C, and the supernatant was transferred to a new tube.

Lysates were incubated with 25-ul slurry (5-uL beads) and rotated for 2 h at 4 °C. Beads were rinsed 3 X with 500 µL Lysis Buffer 300 mM KCl, 0.05% NP-40, 1:1000 DTT, and 1:1000 LPC. Beads were then washed on a rotator at 4 °C 2 × 5 min with 500 µL Lysis Buffer 300 mM KCl, 0.05% NP-40, 1:1000 DTT, and 1:1000 LPC, followed by a single 5 min wash with 500 µL Lysis buffer without detergent.

Immunoprecipitated proteins were eluted 3X with 200 µl 0.1 M Glycine, pH 2.6. Elutions were pooled into a tube containing 100 µl 2 M Tris, pH 8.5. Proteins were then precipitated in 20% trichloroacetic acid (Fisher Bioreagents) overnight on ice. The next day, samples were spun at 20,000×$g$ at 4 °C. Pellets were washed three times with cold acetone and dried in an Eppendorf Vacufuge.

## Quantitative mass spectrometry

Mass spectrometry sample preparation, TMT labeling, peptide fractionation, data acquisition, and analysis were carried out as described in (Ly et al, 2024). Protein abundances were normalized to GFP to account for differences in immunoprecipitation efficiency. Proteins with low coverage (<20%) were excluded from downstream analyses.

## Quantitative PCR assays

Reverse transcription was performed using the Maxima First Strand cDNA Synthesis Kit for RT-qPCR (Thermo Fisher Scientific). cDNA was diluted 1:50 and mixed with 1 µM primers and 2X SYBR Green PCR Master Mix (Thermo Fisher Scientific) in 384-well plates. Three technical replicates were performed per cDNA sample and primer pair. Primer sequences can be found in Table EV2.

## Cell line generation for IAA-induced PABPC1 and PABPC4 degradation

AID was introduced at the C terminus of PABPC4 using Cas9-mediated genome engineering in the PABPC1-AID HCT116 cell line (sC278-C2, Xiang and Bartel, 2021). The 5′ and 3′ homology arms (~500 nt) flanking the stop codon of PABPC4 were amplified from genomic DNA of HCT116 OsTIR1 cells and cloned into the same donor plasmid used to tag PABPC1 (Xiang and Bartel, 2021). The AID coding sequence, followed by a P2A peptide and the hygromycin coding sequence, was inserted immediately before the PABPC4 stop codon. The PAM region targeted by Cas9 on the donor plasmid was mutated. The Cas9 guide RNA was cloned into pX330-BFP, as described (McKinley and Cheeseman, 2014). Both the donor (C287) and the Cas9 (C275) plasmids were co-transfected into the sC278-C2 cell line (plated at $2.5 \times 10^5$ cells/well 24 h before) at 0.6 µg each in a 12-well format with FuGENE HD (Promega). After 48 h, cells were replated with serial dilutions of threefold for three times in 10-cm plates. Four days after transfection, culture media were replaced with fresh media containing Hygromycin B (300 µg/ml). Six days after transfection, culture media were replaced with fresh media containing Hygromycin B (300 µg/ml) and G418 (600 µg/ml), and they were replenished every 4 days. Single colonies were picked using cloning cylinders (Millipore, TR1004), expanded, and genotyped by PCR and western blot. Only clones that were homozygous for AID integration were retained. One clonal line (sC278-287-C8) was used for IAA-induced PABPC1-AID and PABPC4-AID degradation. To minimize background degradation, cells were incubated with 1 µg/ml doxycycline and 0.2 mM auxinole (Aobious, AOB8812), an inhibitor of OsTIR1 (Yesbolatova et al, 2019) for 6 h, after which fresh media with 1 µg/ml doxycycline and 0.5 mM IAA were added for 1 h before cells were harvested for, ting.

## PABP depletion in STLC-arrested cells

HCT116 cells with PABPC1 and 4 AID were kindly gifted by the Bartel lab (Xiang and Bartel, 2021). To deplete PABPC1&4 during mitosis, cells were grown to 20% confluency before 8 µM STLC was added together with 1 µg/ml doxycycline to induce OsTIR1 expression and 0.2 mM auxinole (Aobious, AOB8812) in DMSO to prevent premature OsTIR1 activity. An equivalent volume of DMSO was added to non-induced control cells. After 16 h, fresh media with 8 µM STLC was added to all cells, with 0.5 mM IAA only added to induced cells. For Western blotting, cells were collected at 0, 1, or 4 h after IAA addition. For sequencing, cells were collected as described above at 0 and 4 h. For imaging, SiR-DNA was added (1:10,000), and imaging was started 1 h after the addition of IAA and SiR-DNA.

## Western blotting

Cell pellets were washed once with PBS and resuspended in 1× Laemmli sample buffer (100 mM Tris, pH 6.8, 12.5% glycerol (v/v), 1% SDS (w/v), 0.1% bromophenol blue (w/v), 200 mM β-mercaptoethanol), then boiled at 95 °C for 5 min. Cell lysates were

sonicated at 10% amplitude for 5 s using the Branson Digital Sonifier 450 Cell disrupter to shear genomic DNA. Samples were separated by SDS–PAGE using a 10% gel and transferred to PVDF membrane (VWR). Blots were rinsed once with TBST, then blocked in 5% milk at room temperature for 1 h. Blots were then incubated with primary antibodies diluted in 5% overnight at 4 °C. The next day, the blots were washed for 5 min with TBST 4×, incubated with secondary antibody in 5% milk for 1 h at room temperature, followed by 4x more TBST washes and 1x PBS rinse. Blots were imaged using an Odyssey Clx machine (LI-COR) and analysed with the Image Studio software (LI-COR).

Primary antibodies used in this study were anti-PABP1 (1:1000, Cell Signaling Technology #4992), anti-PABP4 (1:1000, Anti-PABPC4, Novus Biologicals NB100-74594), anti-beta actin (Cell Signaling Technology, D6A8), anti-vinculin (Proteintech 66305), and anti-alpha tubulin (1:5000, Sigma-Aldrich, T9026). Secondary antibodies used in this study were IRDye 680RD Goat anti-Mouse (LI-COR 92668070), IRDye 800CW Goat anti-Rabbit (LI-COR 92632211) at a 1:10,000 dilution.

### Live-cell imaging

For live-cell fluorescence imaging, cells were seeded into 12-well glass-bottomed wells (Cellvis, P12-1.5 P) and grown to 20% confluence. PABPC1 and 4 were depleted as described above. DNA was stained with 1:10,000 SiR-DNA. Images were acquired on a Nikon Eclipse microscope equipped with a sCMOS camera (ORCA-Fusion BT, Hamamatsu) using a Plan Fluor 20×/0.5 NA objective at 10 min intervals. Time-lapse videos were analyzed using FIJI (ImageJ; NIH). STLC-arrested cells were identified by their characteristic monopolar spindles. Slippage was determined as the first frame in which the individual chromosomes were no longer condensed, and timing was calculated using the MTrackJ plugin.

## Data availability

Sequencing data from RNA-seq and PAL-seq experiments is now available on Gene Expression Omnibus: https://www.ncbi.nlm.nih.gov/geo/query/acc.cgi?acc=GSE314020 (GSE314020; RNA-seq) and https://www.ncbi.nlm.nih.gov/geo/query/acc.cgi?acc=GSE314021 (GSE314021; PAL-seq).

The source data of this paper are collected in the following database record: biostudies:S-SCDT-10_1038-S44318-026-00765-5.

## Peer review information

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

## Acknowledgements

This work was supported by grants from the NIGMS (R35GM126930 to IMC). DPB is an investigator of the Howard Hughes Medical Institute. EK is supported in part by the MIT HEALS graduate fellowship. AL is supported by an NIH fellowship (K00CA234921). JL is supported in part by the Natural Sciences and Engineering Research Council of Canada. We thank all the members of the Cheeseman and Bartel laboratories for helpful discussions. We thank the Whitehead Institute Genome Technology Core for the RNA sequencing, the Flow Cytometry Core for assistance with FACS experiments, and the Glass Wash Core for supporting our work.

## Author contributions

**Ekaterina Khalizeva**: Conceptualization; Formal analysis; Investigation; Visualization; Methodology; Writing—original draft; Writing—review and editing. **Arash Latifkar**: Investigation; Methodology; Writing—review and editing. **Kehui Xiang**: Investigation; Methodology; Writing—review and editing. **David P Bartel**: Supervision; Funding acquisition; Writing—review and editing. **Jimmy Ly**: Conceptualization; Funding acquisition; Investigation; Methodology; Writing—review and editing. **Iain M Cheeseman**: Conceptualization; Funding acquisition; Writing—review and editing.

Source data underlying figure panels in this paper may have individual authorship assigned. Where available, figure panel/source data authorship is listed in the following database record: biostudies:S-SCDT-10_1038-S44318-026-00765-5.

## Disclosure and competing interests statement

The authors declare no competing interests.

# Expanded View Figures

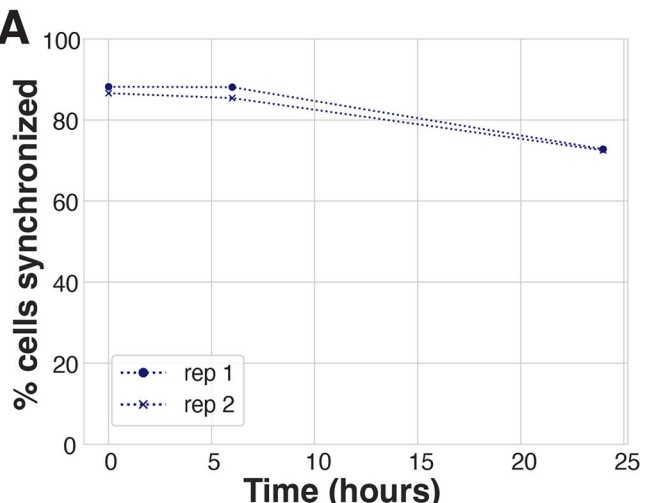

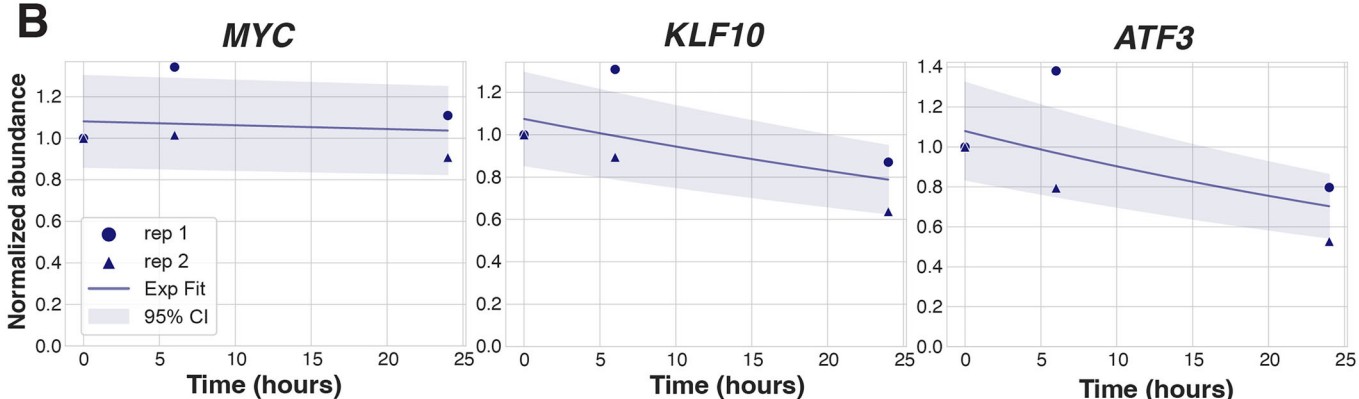

**Figure EV1. Synchronization efficiency and transcript abundances during prolonged mitotic arrest.**

(A) Percentage of cells synchronized in mitosis during each experimental timepoint (Fig. 1), as quantified by FACS analysis of DNA content and pH3(Ser10). (B) Exponential curve fit to individual transcript abundances in STLC-arrested cells, plotted with 95% confidence intervals. *MYC* $t_{1/2}$ 399 h, $R^2$ 0.05. *KLF10* $t_{1/2}$ 54 h, $R^2$ 0.72. *ATF3* $t_{1/2}$ 39 h, $R^2$ 0.78.

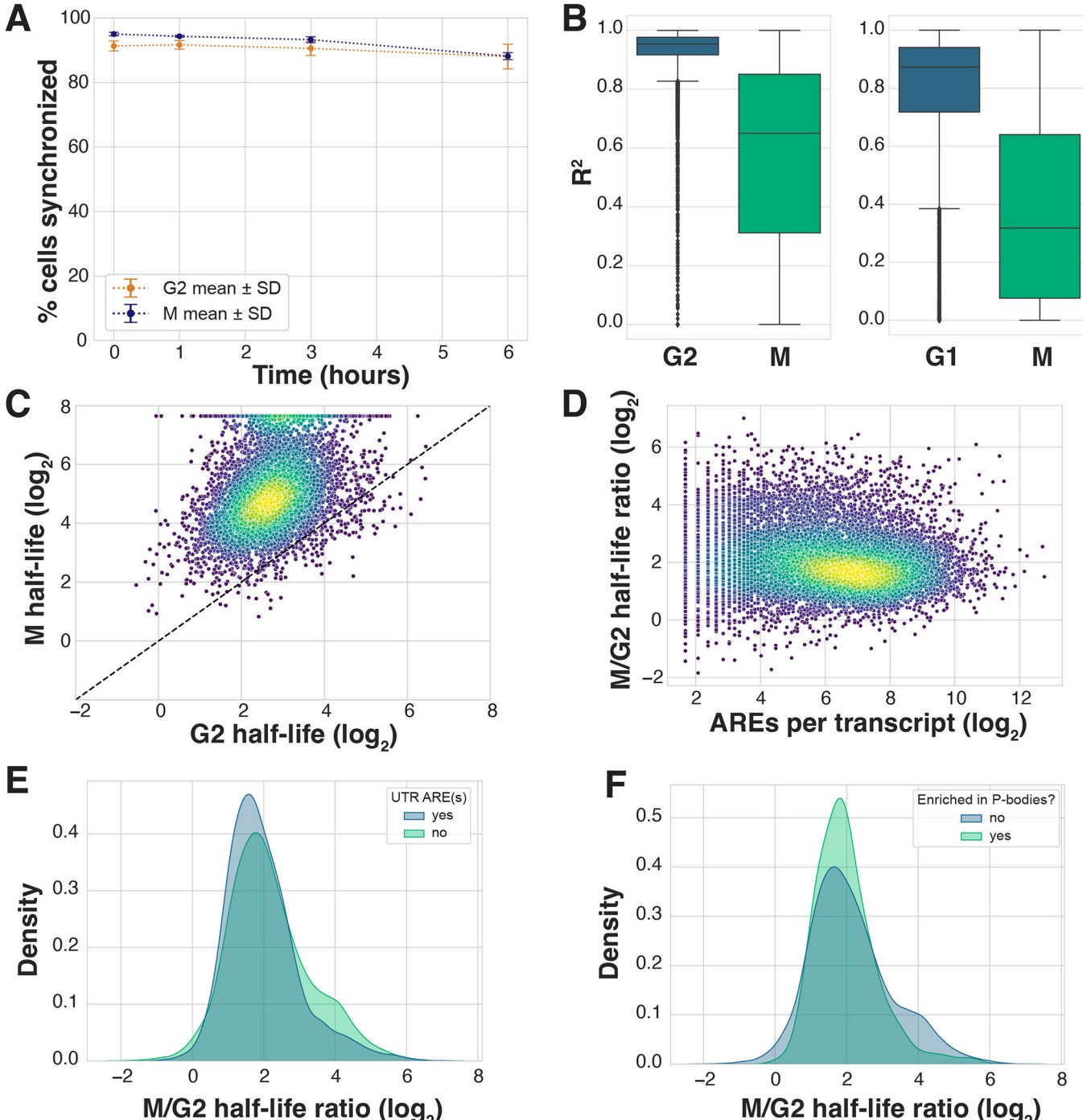

◀  **Figure EV2.  Analyses related to transcription inhibition timecourses.**

(A) Percentage of cells synchronized in G2 or mitosis during each experimental timepoint (Fig. 2), as quantified by FACS analysis of DNA content and pH3(Ser10). (B) Boxplots showing $R^2$ goodness of fit values for half-life calculations for transcripts mapped in transcription inhibition experiments (Fig. 2B, C). Each box shows the interquartile range (IQR) of the data, with the median marked by a horizontal line. The whiskers extend to include the rest of the distribution, except for points that are determined to be outliers based on the IQR. $R^2$ cutoffs were chosen based on interphase fits in order to allow for the possibility of global mRNA stabilization, as a lack of degradation is expected to result in poorer fit values. $n = 10,544$ (left), $n = 10,604$ (right). (C) Scatterplot comparing mRNA half-lives in cells synchronized in G2 or M and treated with THZ1. Data from one biological replicate. $R^2$ cutoff >0.4 was used, $n = 6410$. Median $t_{1/2}$ 6.77 and 33.44 h for G2 and M, respectively. (D) Scatterplot comparing mitotic stabilization and number of AREs per transcript. ARE annotations were acquired from AREsite2 (Fallmann et al, 2016). Arbitrary c = 0.9 was added to all ARE counts to allow log-transformation of the data. $R_S = -0.13$. (E) Probability density function plot comparing the degree of mitotic stabilization among transcripts with or without AREs in the 3′ UTR. 3′ UTR ARE annotations were acquired from https://brp.kfshrc.edu.sa/ared/Home/FTP (Bakheet et al, 2018). $n = 3085$ transcripts with AREs in 3′ UTR, $n = 6773$ transcripts with no AREs in the 3′ UTR. (F) Probability density function plot comparing the degree of mitotic stabilization among transcripts based on their P-body enrichment status (Hubstenberger et al, 2017). P-body enrichment was defined by FDR <0.01 and $\log_2$ enrichment >2, resulting in 1602 P-body-enriched transcripts and 8256 not enriched.

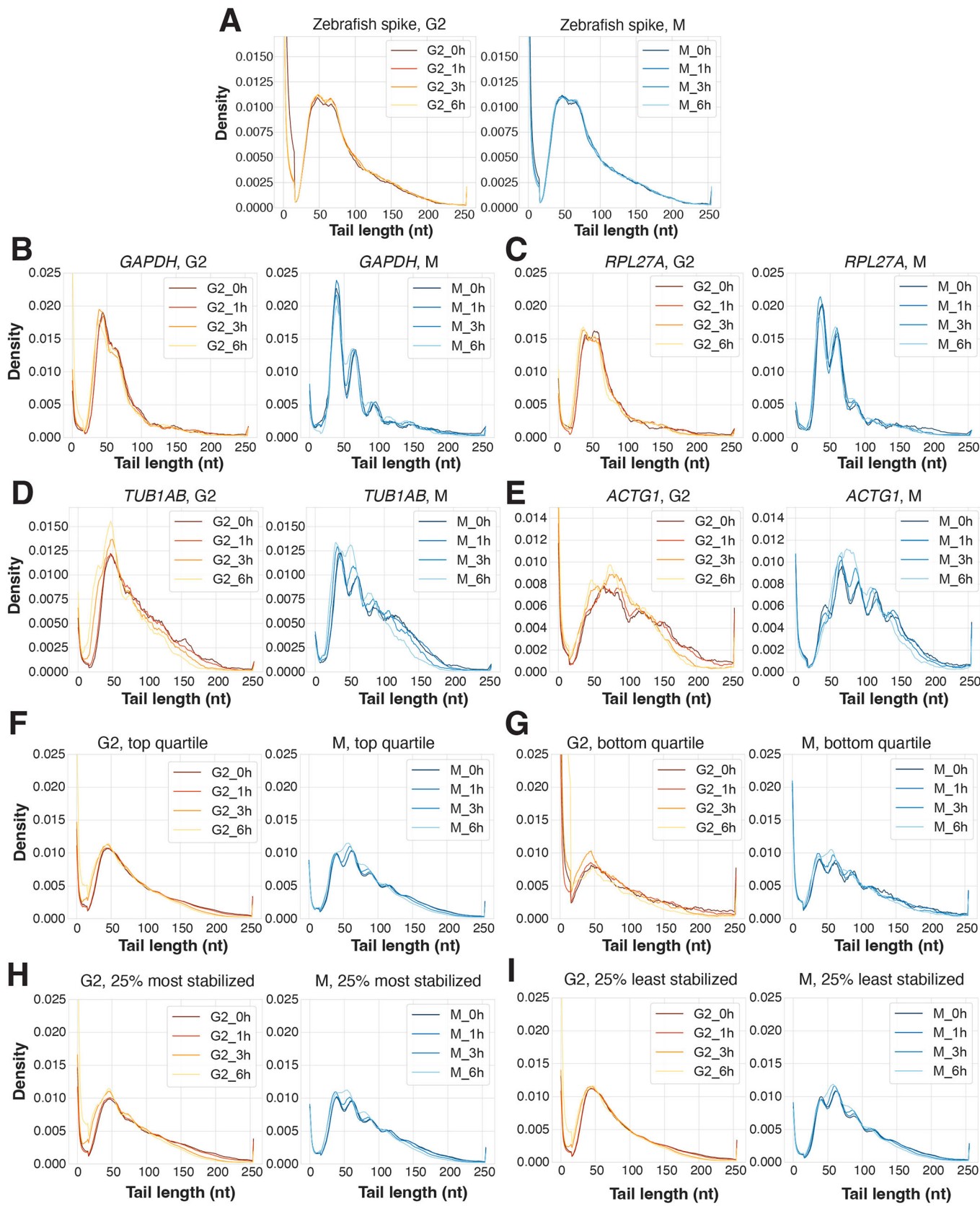

◀  **Figure EV3.  Poly(A) tail-length distributions for subsets of transcripts in G2- and M-arrested cells.**

(**A**) Poly(A) tail-length distributions from zebrafish mRNA spiked into G2- or M-treated samples as measured by PAL-seq. (**B–E**) Poly(A) tail-length distributions in G2- or M-arrested cells treated with actinomycin D as measured by PAL-seq for GAPDH (**B**), RPL27A (**C**), TUB1AB (**D**), ACTG1 (**E**). (**F, G**) Poly(A) tail-length distributions in G2- or M-arrested cells treated with actinomycin D as measured by PAL-seq for 25% most highly expressed genes (**F**) or 25% mostly lowly expressed genes (**G**). (**H, I**) Poly(A) tail-length distributions in G2- or M-arrested cells treated with actinomycin D as measured by PAL-seq for 25% most stabilized transcripts (**H**) or 25% least stabilized transcripts (**I**) based on analysis from Fig. 2.

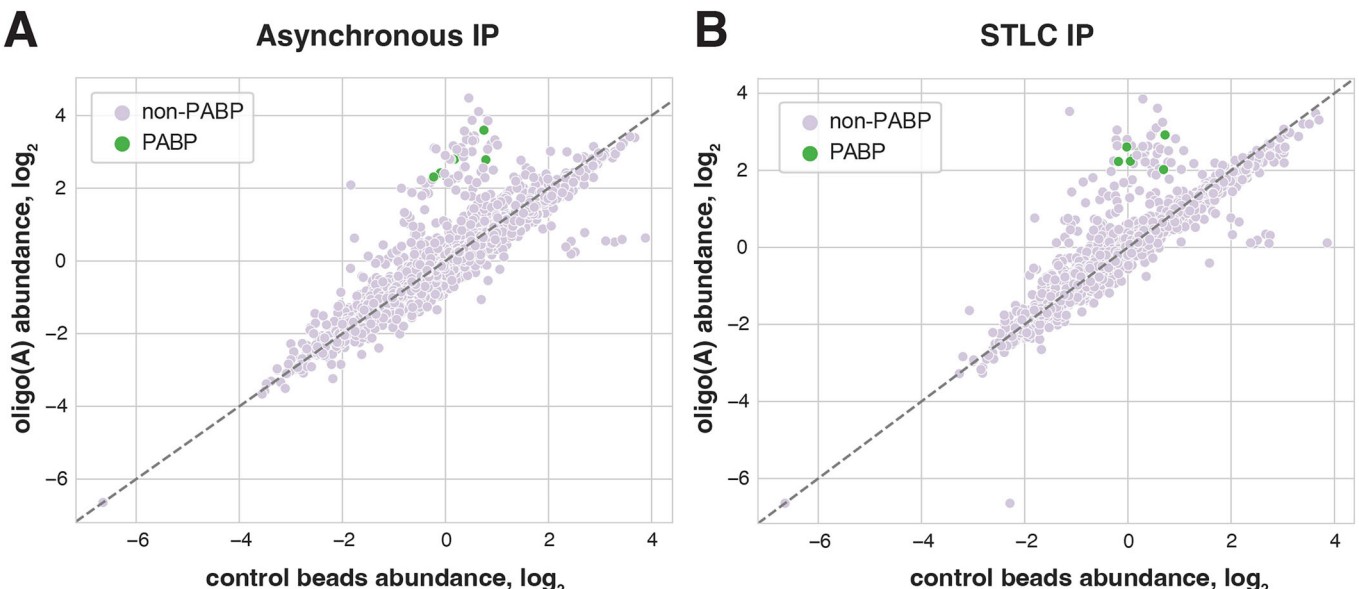

**Figure EV4. Protein abundances in control and oligo(A) pulldowns in asynchronous and STLC-arrested cells.**

(A, B) Scatterplots comparing protein abundance in pulldowns by oligo(A)-coated beads and negative control beads in (A) asynchronous and (B) STLC-arrested cells. Poly(A)-binding proteins are highlighted in green.

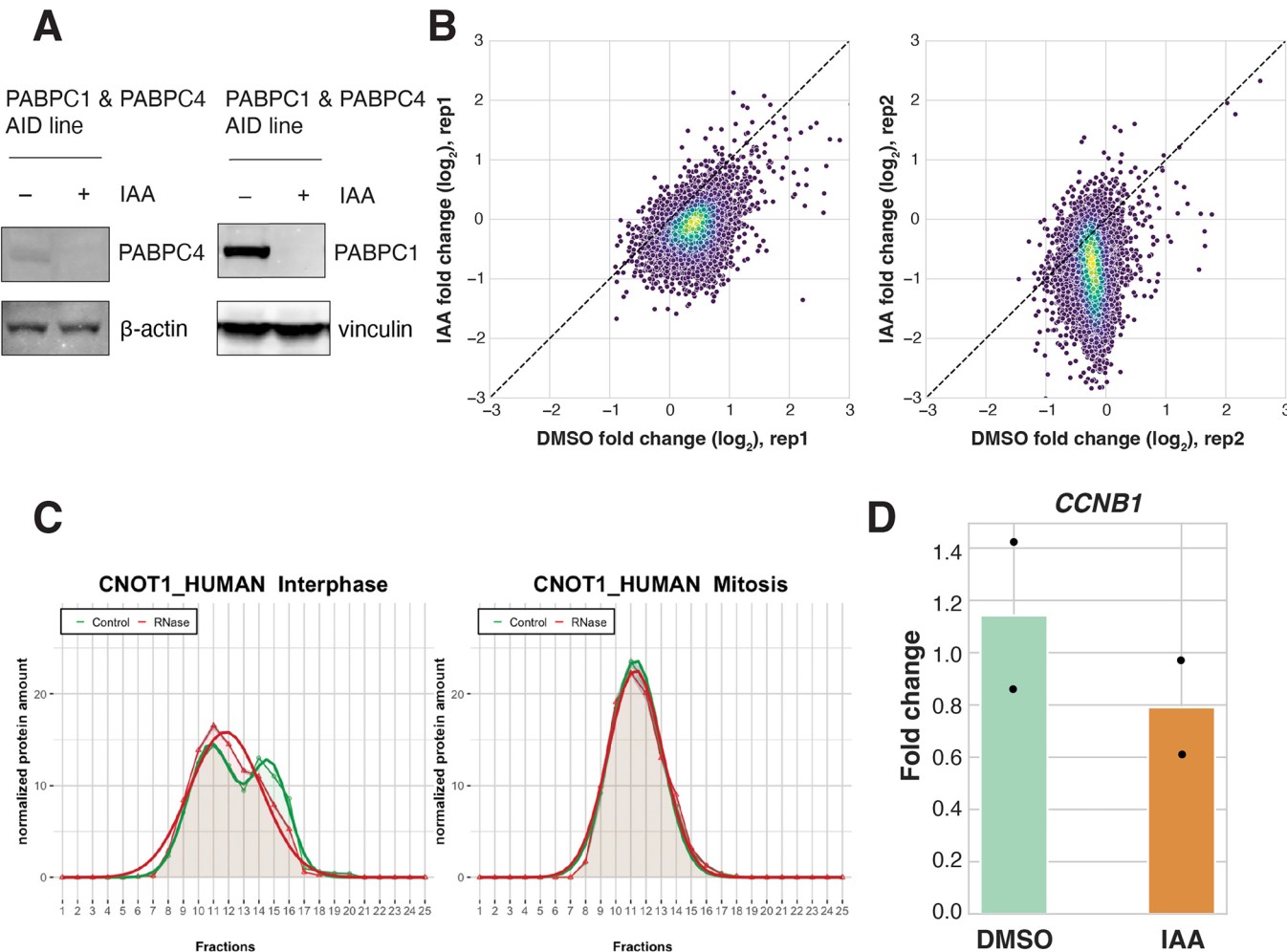

**Figure EV5.  Data related to PABPC1&4 depletion experiments.**

(**A**) Western blots showing PABPC1 & PABPC4 depletion efficiency after 1 h of IAA treatment. Beta-actin and vinculin are shown as loading controls. (**B**) Scatterplots comparing mRNA abundance fold changes in STLC-arrested HCT116 cells within 4 h of either DMSO control treatment or IAA-induced PABPC1&4 depletion. Replicates 1 & 2, $n = 11,078$. (**C**) Graphical representation of the CNOT1 protein amount from interphase or mitotic HeLa cells in 25 different fractions of control (green) and RNase-treated (red) sucrose density gradients analyzed by mass spectrometry. Plots were acquired from https://r-deep3.dkfz.de/ (Rajagopal et al, 2025). The leftward shift of the distribution upon RNase treatment in interphase cells suggests interaction with RNA, whereas a lack of shift in mitotic cells suggests reduced or no interaction. (**D**) Bar graph comparing *CCNB1* degradation in STLC-arrested HCT116 cells within 4 h of either DMSO control treatment or IAA-induced PABPC1&4 depletion. The mean of two biological replicates is plotted; each point represents one biological replicate. Source data are available online for this figure.

