## [Peer Review File · The EMBO Journal]

Global stabilization of the transcriptome in mitotic cells

Ekaterina Khalizeva, Arash Latifkar, Kehui Xiang, David Bartel, Jimmy Ly, and Iain Cheeseman

Corresponding author(s): Iain Cheeseman (icheese@wi.mit.edu) , Jimmy Ly (jimmyly@wi.mit.edu)

Review Timeline:

Submission Date:	21st Jul 25
Editorial Decision:	2nd Sep 25
Appeal Received:	29th Jan 26
Editorial Decision:	26th Feb 26
Revision Received:	15th Mar 26
Accepted:	17th Mar 26

Editor: Hartmut Vodermaier

Transaction Report:

Prof. Iain M Cheeseman
Whitehead Institute for Biomedical Research
Massachusetts Institute of Technology
Department of Biology
Suite 401, 455 Main St.
Cambridge, MA 02142

2nd Sep 2025

Re: EMBOJ-2025-121943
Global inhibition of deadenylation stabilizes the transcriptome in mitotic cells

Dear Iain,

Thank you for submitting your manuscript on transcriptome stabilization during mitotic arrest to The EMBO Journal. I apologize for the delay in its evaluation due to the summer vacation period affecting referee availability as well as staffing of our editorial office. We have now received the reports of three expert referees, and we have also been able to discuss them in detail within our team. I am afraid to say our conclusion was that we presently cannot offer to publish this study.

As you will see from the comments below, while referee 1 offers some brief supportive comments, referees 2 and 3 are at this stage not in favor of EMBO Journal publication. Both of them clearly acknowledge the importance of the question and the potential interest of your analyses, including the characterization of global mRNA stabilization during mitosis. However, neither of them was convinced that the data provide sufficiently decisive evidence for this being caused by mitotic repression of mRNA deadenylation. Since their arguments for this are explained in their well-argued reports, I prefer not to repeat them in detail here in this letter; but hope you understand that these shared reservations affecting the key novel conclusion of the study preclude us from inviting (and thus to some degree committing to) a revised version of this manuscript.

Should future work allow you to strengthen the support for deadenylation inhibition during mitosis being causal for reduced mRNA turnover, I would be happy to once more look into the study and to consider the possibility of sending a resubmission back to the original referees. In this case, I would however encourage you to first contact me with preliminary responses to the reviewers' reports, so we could gauge the potential chances during re-review ahead of resubmission. Alternatively, I would be happy to explore the option of transferring a rewritten/reorganized manuscript -focusing on the general characterization of mRNA half-life increase during mitosis and toning down the conclusions on this being due to deadenylation- with the editors of our sister journal EMBO Reports; please do let me know in case you should be interested in this possibility.

I am sorry that the referee reports do not allow me to be more positive for The EMBO Journal at the current stage, but hope that you will nevertheless find the referees' comments and suggestions helpful when considering how to proceed further with this work. Thank you once more for having had the opportunity to consider this work for publication.

With kind regards,

Hartmut

Referee #1:

This study demonstrates that mRNA levels are maintained during mitosis despite a stoppage of transcription due to mRNA stabilization that is achieved by a reduction in deadenylation that requires PABPC, likely through a decrease in CCR4-NOT activity. This observation extends our understanding of transcriptome buffering phenomena and builds upon and extends previous observation of waves of mRNA decay during the cell cycle. Overall the experimental design is appropriate and the data provide solid support for the conclusions that are drawn. I only have a few minor suggestions for polishing the manuscript.

Minor Points:

1. Fig. 1/2: The out of sequence presentation of panels (panel E in Fig. 1; panel D in Fig. 2) in the middle of figure is suboptimal for readers. I would recommend realigning the figure panels for better flow.
2. Given the highlighting of the interesting UBE2C 'exception to the rule' data in the Results section, I might suggest moving these data from extended view into the main set of figures.

Referee #2:

In this paper, Khalizeva, Ly, Cheeseman and colleagues address how cells arrested in mitosis maintain their mRNA pool despite the absence of measurable transcription. This is a fundamental question, and it may have relevance in cancer biology.

The study first shows that mRNAs are indeed maintained for 6 hours and only moderately reduced after 24 hours of mitotic arrest. The authors then compare mRNA half-lives and poly(A) tail length between mitotically arrested and G2-arrested cells. They find the mRNA half-lives in mitotically arrested cells prolonged and observe differences in poly(A) tail length. The significance of the poly(A) tail length differences remains unclear, at least in my opinion. I appreciate that the authors used a second transcription inhibitor, THZ1 (Fig. EV2C), in addition to actinomycin D (Fig. 2), which provides additional confidence in the mRNA half-life measurements.

The authors then demonstrate that removal of the poly(A) binding PABPC1&4 destabilizes mRNAs of cells arrested in mitosis (presumably it would also destabilize interphase mRNAs, but this has not been tested here). This mRNA destabilization coincides with slippage out of mitosis, from which the authors conclude that maintaining stable mRNAs during mitosis is functionally important.

Several papers have described differences in mRNA degradation kinetics across the cell cycle, but they were typically focused on specific transcripts, periodic changes in degradation, or a different phase of the cell cycle. To my knowledge, none has addressed the precise question asked here. Poly(A) tail changes across the cell cycle have been examined as well, and I think this deserves a more thorough discussion in the paper (see below).

Overall, the experiments are appropriate and informative, but I feel that the reasons for the longer mRNA half-lives during a mitotic arrest remain unknown (they may well be multi-factorial) and that the authors' hypothesis of mitotic repression of deadenylation is poorly supported. The results, as they are, are well presented and the paper is concisely written. The scope of the study is in principle appropriate for EMBO J, but I think the authors either need to support their conclusions with additional experiments, or need to tone down their conclusions.

Major comments:

(1) I have a technical concern regarding how the normalization is done. mRNA reads are normalized to total RNA (spike-in is a defined amount per total RNA). This makes the assumption that total RNA per cell remains constant (in a prolonged arrest, or during drug treatment). If this assumption were wrong, the mRNA levels would be misinterpreted. (For example, if total RNA levels declined, the authors may plot mitotic mRNAs as stable (Fig. 1), when they actually decline along with total RNA.) Or maybe I am misunderstanding what was being done. If so, a more detailed explanation, separated by the type of experiment, would be helpful.

(2) I do not think the data strongly support a mitotic repression of deadenylation.

In my opinion, the authors jump to this conclusion prematurely. One of the first sentences about the poly(A) tail lengths result is: "the tail length distributions in mitotic cells appeared markedly different from those of G2-arrested cells, suggesting repressed deadenylation during mitosis" (Page 7). While I fully agree with the first part of the sentence, I don't think the second part is supported. The median poly(A) tail lengths between G2 and M, as measured here, are highly similar (72 vs. 76 nt, both with a peak just under 50 nt and stretching out to 250 nt, Fig. 3B/C/D). Furthermore, mRNAs that were less or more stabilized in M-arrested cells don't show obvious differences in their poly(A) tail length profile (Fig. EV3H/I).

The authors base their conclusion on a very small difference in the 10 - 25 nt range after treatment with actinomycin D (bottom left panels in Fig. 3B/C), from which they want to conclude that "shorter-tailed transcripts are not deadenylated as efficiently during mitosis as they are in interphase." The authors mention the alternative conclusion that "short-tailed isoforms are more efficiently cleared in mitotic cells," but discard it because they say it is inconsistent with the mRNA stabilization in mitosis. However, since this seems to concern only a very minor pool of total mRNA, I am not sure this would necessarily be reflected in the mRNA half-lives. Also, this difference only appears after actinomycin D treatment. Any difference that is functionally important between G2 and M should be obvious prior to any additional drug treatment.

In fact, I do not fully understand why the authors felt that poly(A) tail length in G2 versus M needed to be examined after transcriptional inhibition (Fig. 2B/C). Presumably to somehow get the kinetics. But M-phase arrested cells are already transcriptionally blocked (Fig. 1E). So this seems an unequal comparison. I'd appreciate it if the authors could explain their rationale.

The largest difference between G2 and M poly(A) profiles are the "phased PABPC toeprints" obvious in M-arrest but not G2-arrest (Fig. 3B/C). I think a deeper exploration and discussion of this difference would be useful.

The authors give the impression that these toeprints are not normally observed and only seen after transcription inhibition (Eisen et al. 2020) or inactivation of CCR4-NOT (Yi et al., 2018). However, in my opinion, the Yi et al. paper already shows toeprints in control RNAi (Fig. 1F/G) and they are actually rather reduced in CCR4-NOT knock-down (unless I read their data wrongly). Furthermore, when Khalizeva et al. inhibited transcription in G2-arrested cells, this didn't make toeprints appear (Fig. 3B). So, it remains unclear to me what causes them. At least one transcript shows toeprints in G2 (Fig. EV4C). Can anything be learned from which transcripts do and which transcripts don't? Do any mRNAs in M-phase arrest not show toeprints?

I also don't agree that the "striking phasing of PABPC toeprints.... suggest that PABPC is not efficiently removed from the poly(A) tails" in mitosis (page 8). It is entirely possible that PABPC is equally strongly bound in G2 and M, but that poly(A) tails have more flexible lengths beyond the PABPC-protected region in G2. It needs other types of measurements to demonstrate that there is less PABPC associated with G2 than with M transcripts or that PABPC is more strongly bound in M-phase arrested cells.

Overall, the most plausible interpretation is that some unidentified aspect of mRNA degradation is affected in M-phase arrested cells, and this may secondarily cause the phased poly(A) length profile. However, a global repression of deadenylation is not obvious in M- versus G2-arrested cells in my opinion. (Compare the G2/M difference here to how strongly poly(A) tail lengths shift when CCR4-NOT is knocked down in Yi et al., 2018.) I feel the authors either need to provide stronger evidence or tone down their conclusion.

(3) Similarly, I don't think it is justified to say that PABPC stabilizes transcripts specifically in mitosis (page 9). The PABC1&4-depletion experiment doesn't allow this conclusion. Yes, PABC1&4 depletion destabilizes mRNAs in the mitotic arrest. However, G2 was not tested. And Xiang and Bartel, eLife 2021, show that mRNA half-lives in asynchronously growing HeLa cells were reduced after PABPC knockdown. So whether this effect of PABPC on mRNA stability contributes to the enhanced mRNA stability in a mitotic arrest is unclear.

(4) The authors observe mitotic slippage in cells depleted of PABC1&4 using the AID system. This experiment would benefit from a control where cells are treated with IAA without the presence of the AID tags to demonstrate specificity. In addition, since the authors want to conclude that destabilization of the mRNAs leads to slippage, can they show that slippage is rescued by blocking mRNA degradation (maybe by inhibiting decapping)? Without any such evidence, wouldn't it be possible that PABC1&4 depletion impairs translation, and this leads to slippage?

(5) Although I appreciate the conciseness of the text, a little more information would occasionally be helpful. For example, the main text would benefit from information on the length of drug treatments or how cells were synchronized in G1 (release from mitotic arrest).

(6) I feel the prior literature could be more thoroughly and more accurately covered. My suggestions below are not comprehensive, just some examples.

A surprising omission in the references is Park Mol Cell 2016 (<https://doi.org/10.1016/j.molcel.2016.04.007>) where poly(A) tail lengths in S and M phase were measured. Other examples are: Novais-Cruz et al., eLife 2018 (<https://doi.org/10.7554/eLife.36898>), and Sloss et al., Oncotarget 2016 (<https://www.oncotarget.com/article/6894/>). Krenning et al., eLife 2022, is relevant and is indeed cited, but could be discussed more thoroughly. For example, they also show stable mRNAs in a mitotic arrest. On page 2/3 "Following the resolution of any mitotic damage, ...": Sinha et al., Cell Cycle 2019, does not seem an ideal citation here since it mostly deals with "slippage", i.e. not resolution, and is focused on drug-resistance in cancer.

(7) The Xiang and Bartel, eLife 2021 paper that is cited for PABC1&4 AID only reports PABC1-AID. Please provide the missing information on PABC4-AID.

(8) I did not see any information on data availability.

(9) The authors occasionally pick single transcripts to give examples, e.g. MYC, KLF10, ATF3 in Fig. 2 or UBE2C in Fig. EV4. I wonder whether the data could be additionally mined for interesting, gene-specific information. For example:

(1) The M/G2 half-life ratios for coding and non-coding RNAs have a distinct shoulder or second peak (Fig. 2F). What is the identity and nature of these RNAs with a considerably higher ratio?

(2) Clearly, at least one RNA shows distinct PABPC toeprints in G2 (Fig. EV4). Fig. 3B shows the average of thousands of RNAs. Which fraction of them shows toeprints? And what is the nature of the ones that do?

Minor comments:

- What is the reason for the R² cutoff in the mRNA half-life experiments? The explanation given ("in order to allow for the

- possibility of global mRNA stabilization") is not sufficiently clear. What fraction of genes was removed? This information is important, as the filter may change the estimate of the median mRNA half-life across the genome.
- PAL-seq experiment: How many replicates? Over 6 hours the median tail length in G2 decreased by 14 nt and in mitotic arrest by 9 nt. Because these numbers are fairly small, it would be useful confirm that they are not the result of random sampling effects with some statistical testing.
 - Everything in this paper deals with a mitotic arrest situation, not with unperturbed mitosis. I think it is important to consistently use "mitotic arrest" phrasing in the conclusions and avoid giving the impression that the results apply to "mitosis" in general - although it is of course valid to speculate that they might.
 - Maybe highlight the genes shown in Fig. 2C in the Fig. 2B and 2D graphs - just to illustrate where they fall on the spectrum.
 - Why does 2E show the M/G1 ratio when M/G2 is shown in all other graphs? Show M/G2 in addition?
 - Abstract "... show that mRNA stabilization in mitosis is dependent on ... PABPC1&4". While this statement is correct, it gives the impression that this is a mitosis-specific effect, which it very likely is not (also see above). If I am not mistaken, Xiang and Bartel, eLife 2021, showed a decreased mRNA half-life in asynchronously growing cells.
 - Page 2, "downregulated transcription, as evidenced by chromosome condensation". This seems a little misleading since downregulated transcription does not require chromosome condensation (Zhao et al., Nat Genetics 2024, 10.1038/s41588-024-01759-x, or Spencer et al., JCB 2000, <https://doi.org/10.1083/jcb.150.1.13>)
 - Page 4, "very close to their initial levels even after 24 hours of mitotic arrest". Please put a number on this. The graph is in log₂ scale, which makes any changes look a little smaller than they are. Maybe report the median fold change between each time point. Similarly, on page 11 "maintained over a 24-hour mitotic arrest" isn't quite accurate, since mRNA levels go down to about 70 % after 24 hours.
 - Page 5 "no correlation between the changes in mRNA half-life and translation efficiency" might be too strong for a correlation coefficient of -0.12 (small but not extremely small). "no strong correlation" would work better in my opinion.
 - Page 6, typo in "deadynylases".
 - Page 10 "range of fold-change values was also wider..., consistent with restored regulation..." I do not understand the authors' conclusion. Could they rephrase?
 - Page 12 "leads to a strong reduction mRNA stability" should be "reduction in mRNA stability".
 - Page 12 "the effect of PABPC1&4 depletion... likely to involve ... stabilizing" Shouldn't this be "destabilizing" since the sentence talks about the AID experiment?
 - Page 15 "Cells were washed with 1mL cold PBS and permeabilized with 1mL of 3.7% formaldehyde" should probably read "fixed with 1 mL of 3.7 % formaldehyde..."
 - Methods paragraph on half-life calculations: "mRNA read counts were normalized to spike-in or mitochondrial RNA read counts in pharmacologically transcriptionally inhibited and uninhibited cells, respectively." Not quite correct. Fig 1 experiment does not use transcription inhibition but does use spike-ins.
 - Fig. 3 legend, re-phrase "Insets..."
 - Fig. 3B-C legend: should read "distributions in (B) G2- or (C) M-arrested cells..."
 - Legend to Fig. EV3 uses both "G2- or M-arrested" and "RO-3306- or STLC-arrested". Maybe make the phrasing consistent.
 - All the information from Fig. 4D is also obvious in Fig. 4C. What is the benefit of the additional plot?
 - What do the error bars in Fig. EV1A and EV1B represent? If these are from the two replicates, rather show the individual points from the two replicates.
 - What do the error bars in Fig. EV5C represent?

Referee #3:

In this paper by Khalizeva and colleagues, the authors study mRNA stability during mitosis. The authors first show that mRNA turnover is greatly reduced in mitotic cells. They then go on to measure polyA tail length and identify differences in polyA tail length distributions in mitotic vs interphase cells. They further show that rapid depletion of polyA binding proteins PABPC1/4 results in increased mRNA degradation during mitosis. Finally, the authors go on to show that increasing global mRNA turnover during mitosis through depletion of PABPC1/4 results in a hampered ability of cells to maintain a prolonged mitotic checkpoint arrest. Together, the authors then propose a model in which deadenylation of polyA tails is reduced in mitosis, leading to a reduced mRNA turnover, which is important for maintenance of protein synthesis during a prolonged mitotic arrest. Overall, the paper was well-written and that data clearly presented. The conclusion that mRNA stability was altered in mitosis was well supported by the data, but I was not entirely convinced that the authors have shown conclusively that reduced mRNA turnover was a consequence of a mitotic inhibition of deadenylation. The authors provide some measurements related to polyA tails and perform perturbations to the polyA binding proteins PABPC, but the evidence that deadenylation is reduced in mitosis is indirect at best, and, in my view, currently not strong enough to warrant the strong conclusion made in this study (even in the title); that altered deadenylation is the cause of altered mRNA stability in mitosis. If the authors can provide direct proof that deadenylation is inhibited during mitosis and that this inhibition causes the reduced mRNA turnover, then I think this paper would be well suited for publication.

I found the logic in the section describing figure 3 hard to follow. It wasn't clear to me how phasing of PABPC toeprints would represent proof of reduced deadenylation activity in mitosis.

The authors deplete PABPC1/4 and observe that mRNA becomes less stable in mitosis and conclude that: "This result suggests that the global transcriptome stabilization observed in mitotically arrested cells occurs through the inhibition of deadenylation."

I don't see how this conclusion can be reached from this experiment. PABPC1/4 are well known mRNA stabilizing protein, depleting them is expected to cause a decreased mRNA stability irrespective of the underlying mechanism that leads to reduced mRNA turnover in mitosis.

Moreover, if deadenylation was globally inhibited in mitosis, why are mRNAs rapidly degraded upon PABPC1/4 removal (as such decay likely involves deadenylation)? The experiment rather suggests that the deadenylation machinery is still active in mitosis (although this experiment doesn't really provide a quantitative analysis of deadenylation activity in mitosis vs interphase, it just shows that at least some deadenylation activity is present in mitosis).

The authors show that polyA tails are slightly longer in mitosis than in interphase (~10-20%), but it is not clear whether the difference in polyA tail length is large enough to explain the much larger difference in mRNA stability (3-4 fold). Moreover, no clear evidence is provided that show that the slight increase in polyA length in mitosis is in fact causative for the altered mRNA stability.

Minor point:

In the introduction the authors cite a median mRNA half-life of 2-4 h, this seems on the short side, could the authors check additional literature references?

*** As a service to authors, The EMBO Journal offers the possibility to directly transfer declined manuscripts to another EMBO Press title (EMBO Reports, EMBO Molecular Medicine, Molecular Systems Biology) or to the open access journal Life Science Alliance launched in partnership between EMBO Press, Rockefeller University Press and Cold Spring Harbor Laboratory Press. The full manuscript (including reviewer comments, where applicable and if chosen) will be automatically forwarded to the receiving journal, to allow for fast handling and a prompt decision on your manuscript. For more details of this service, and to transfer your manuscript to another EMBO title please follow this link:

Link Not Available

We thank the reviewers for their constructive comments and suggestions. For the revised manuscript, we have made substantial changes to the text and figures in response to their suggestions. In particular, we have added new experiments to evaluate the contributions of multiple steps of the mRNA degradation pathway to mitotic stabilization of the transcriptome, in addition to deadenylation. We have also used caution with the wording of our conclusions regarding the role of deadenylation in this process. The changes we have made include:

1. Demonstrating that siRNA-mediated mRNA decay is active in mitotic cells.

Although our work implicates changes in deadenylation as a key factor in mitotic mRNA stabilization, the reviewers highlighted that additional aspects of mRNA decay may be affected. To evaluate downstream steps in mRNA degradation, we have now used siRNAs to induce endonuclease-mediated cleavage of a reporter transcript, bypassing the upstream processes of deadenylation and decapping. We found that siRNA-mediated decay remains active during mitosis, suggesting that endonuclease-mediated cleavage and the subsequent degradation of the cleavage products by XRN1 and the exosome can occur in mitotic cells. Thus, the stabilization of mitotic mRNAs likely occurs primarily at upstream steps, including by inhibiting deadenylation.

- 2. Evaluating the contributions of deadenylases to mitotic mRNA decay.** Our prior results highlighted an important contribution of inhibiting deadenylation to stabilizing mRNAs during mitosis. However, as the reviewers rightly pointed out, the mechanism of this phenomenon is not entirely clear. For the revised manuscript, we conducted new experiments to address the possibility that recruitment of deadenylation machinery to transcripts may be impaired. Using a reporter-based strategy to tether factors to an mRNA, we find that tethering the CNOT6 and CNOT7 deadenylases can induce enhanced reporter mRNA degradation in interphase, but

not in mitotically arrested cells. This suggests that merely recruiting these factors to a target transcript is not sufficient to override mitotic inhibition of mRNA decay.

3. **Proteomics of CCR4–NOT and oligo(A)-interacting proteins.** In addition to the functional experiments described above, we have evaluated changes in protein-protein and protein-RNA interactions in mitotic cells. First, we tested the CCR4–NOT deadenylase complex interactions using immunoprecipitation of ectopically expressed CNOT1-GFP followed by quantitative mass spectrometry. We found that the CCR4–NOT complex composition and interactors are not significantly different in interphase and mitotic cells. Additionally, based on pull-downs using oligo(A)-coated beads followed by mass spectrometry, we did not identify differences in oligo(A) protein binding in interphase and mitotic lysates. These experiments demonstrate that altered CCR4–NOT complex composition is unlikely to be the mechanism for the observed reduction in deadenylation, suggesting that other mechanisms are responsible for its altered activity.

4. **PABPC1&4-AID cell line generation and characterization.** Although a PABPC1-AID cell line had been described previously, the double tagged cell line is new for this paper. We have now provided a detailed description of how the PABPC1&4-AID double degron cell line was generated, as well as experiments evaluating the AID efficiency for both proteins.

PABPC1 & PABPC4
AID-degron line

5. **Clarifying the contribution of PABPC1&4 to mitotic mRNA stabilization.** We have clarified our rationale for emphasizing the role of PABPC1&4 in mitotic mRNA stabilization. We agree with the reviewers that a key function of cytoplasmic PABP is to stabilize transcripts in a range of conditions, including during interphase. However, the fact that that PABPC1&4 co-depletion can “override” the inhibition of the mRNA degradation pathway that occurs in mitotic cells (Fig. 2) provides critical information about which steps in the mRNA degradation pathway are attenuated. If both deadenylation and the downstream steps in the mRNA decay pathway were completely inactivated, even PABPC1&4 depletion would not have resulted in a decrease in mRNA stability. We have clarified this logic in the manuscript, emphasizing the implications of the PABP depletion experiments for the state of mRNA degradation machinery in mitosis.

Below we include detailed point-by-point responses to each reviewer comment.

Referee #1:

1. *Fig. 1/2: The out of sequence presentation of panels (panel E in Fig. 1; panel D in Fig. 2) in the middle of figure is suboptimal for readers. I would recommend realigning the figure panels for better flow.*

We have now reorganized these panels as suggested.

2. *Given the highlighting of the interesting UBE2C 'exception to the rule' data in the Results section, I might suggest moving these data from extended view into the main set of figures.*

We thank the reviewer for this suggestion. We have moved this data to the main figures as recommended by this reviewer.

Referee #2:

Overall, the experiments are appropriate and informative, but I feel that the reasons for the longer mRNA half-lives during a mitotic arrest remain unknown (they may well be multi-factorial) and that the authors' hypothesis of mitotic repression of deadenylation is poorly supported. The results, as they are, are well presented and the paper is concisely written. The scope of the study is in principle appropriate for EMBO J, but I think the authors either need to support their conclusions with additional experiments, or need to tone down their conclusions.

We thank the reviewer for these comments. For this revised paper, as described throughout this response, we have provided additional data to evaluate other steps in the mRNA degradation process. For example, we now show that siRNA-mediated transcript degradation is still active in mitotic cells, indicating that exonucleases are not inactivated during mitosis. We have also updated our wording and conclusions regarding the contribution of altered deadenylation to mitotic mRNA stability.

(1) I have a technical concern regarding how the normalization is done. mRNA reads are normalized to total RNA (spike-in is a defined amount per total RNA). This makes the assumption that total RNA per cell remains constant (in a prolonged arrest, or during drug treatment). If this assumption were wrong, the mRNA levels would be misinterpreted. (For example, if total RNA levels declined, the authors may plot mitotic mRNAs as stable (Fig. 1), when they actually decline along with total RNA.). Or maybe I am misunderstanding what was being done. If so, a more detailed explanation, separated by the type of experiment, would be helpful.

This reviewer is correct that our normalization for these experiments assumes that overall RNA levels remain similar during the time course in mitotic cells. Importantly, for these experiments, we are measuring **total cellular RNA**, not just mRNA, to calculate the appropriate amount of spike-in for each sample. Given the fact that >95% of total cellular RNA corresponds to ribosomal and transfer RNA and given the well established long rRNA and tRNA half-lives (on the order of days, PMID 19696118, 6628827, 5954370, 29463900), we believe that it is reasonable to assume that total RNA levels remain unchanged. In our experiments, the cells are arrested and therefore not dividing, which also means that the rRNA does not undergo dilution due to cell growth, further preventing rRNA levels from decreasing. We have considered other ways of normalizing the RNA-seq data, but believe that those are more problematic. For example, standard methods of normalization such as TPM or RPKM rely on normalizing to **total mRNA**, which would mask global changes in mRNA levels as this reviewer indicates. Normalization to spiked-in RNA is widespread as a method to approximate absolute quantification in sequencing experiments (PMID 29247756, 33204768, 34159316, 21816910, and many others). For the revised manuscript, we have now added an improved explanation of this reasoning to the methods section to make this important point clearer.

(2) I do not think the data strongly support a mitotic repression of deadenylation. In my opinion, the authors jump to this conclusion prematurely. One of the first sentences about the poly(A) tail lengths result is: "the tail length distributions in mitotic cells appeared markedly different from those of G2-arrested cells, suggesting repressed deadenylation during mitosis" (Page 7). While I fully agree with the first part of the sentence, I don't think the second part is supported. The median poly(A) tail lengths between G2 and M, as measured here, are highly similar (72 vs. 76 nt, both with a peak just under 50 nt and stretching out to 250 nt, Fig. 3B/C/D). Furthermore, mRNAs that were less or more stabilized in M-arrested cells don't show obvious differences in their poly(A) tail length profile (Fig. EV3H/I).

We agree with the reviewer that, at the initial timepoint, the median poly(A)-tail lengths are quite similar between G2- and M-arrested cells. This is expected given the fact that the M-arrested cells had only recently entered mitosis, and the G2 cells had not yet undergone transcription inhibition. Therefore, both conditions at this timepoint reflect what is essentially a steady-state poly(A)-tail length distribution. However, we found it notable that, after 6 hours of transcriptional inhibition, a difference between median tail lengths does become apparent (58 nt in G2, compared to 67 nt in mitotic cells). Although not drastic, this observation is consistent with other data that indicate a reduction in deadenylation during mitosis, such as the distinctive PABPC toeprints and data presented in Fig. 4. We have clarified this point in the main text.

The authors base their conclusion on a very small difference in the 10 - 25 nt range after treatment with actinomycin D (bottom left panels in Fig. 3B/C), from which they want to conclude that "shorter-tailed transcripts are not deadenylated as efficiently during mitosis as they are in interphase." The authors mention the alternative conclusion that "short-tailed isoforms are more efficiently cleared in mitotic cells," but discard it because they say it is inconsistent with the mRNA stabilization in mitosis. However, since this seems to concern only a very minor pool of total mRNA, I am not sure this would necessarily be reflected in the mRNA half-lives.

As most mRNA degradation happens on transcripts whose poly(A)-tails have been shortened, degradation of short-tailed species makes the predominant contribution to a transcript's half-life. In the mitotic time course, we observe a smaller leftward shift of the poly(A) tail length distribution as compared to G2 time course (Fig. 4B-C). There could formally be two explanations for this: 1) during mitosis tails are not shortened as efficiently over time as during G2; and 2) during mitosis poly(A) tails are getting shortened as efficiently as during G2, but instead get degraded more rapidly, thus not contributing to the poly(A) tail length distribution. We propose the second scenario as unlikely, as continued deadenylation paired with efficient clearance of short-tailed mRNA species is precisely the mechanism determining mRNA half-lives in interphase cells. If explanation (2) were the case, mitotic half-lives would be comparable to those in interphase, which would contradict our finding of increased mitotic mRNA stability (Fig. 2). We therefore find it implausible that short-tailed isoforms would be efficiently cleared in mitosis given the global mRNA stability we observed, and conclude that

the first possible explanation of less efficient deadenylation is the more likely scenario.

Also, this difference only appears after actinomycin D treatment. Any difference that is functionally important between G2 and M should be obvious prior to any additional drug treatment.

Figure 4D demonstrates that, without actinomycin D, poly(A) tail length distributions in mitotic cells are markedly different from those in G2 cells, with striking PABPC toeprints arising rapidly upon entry into mitosis.

For experiments comparing the behavior of G2 and mitotic cells, it is important to consider that mitotic cells lack new transcription (Fig. 1C), creating a distinct starting point. In G2 cells, active transcription allows cells to constantly replenish longer-tailed mRNA species as tails get shorter due to continuous deadenylation. In contrast, in mitotic cells, transcription is inhibited. If deadenylation was still active in these cells, the steady-state poly(A)-tail length distribution would indeed have been different from the G2 steady-state. However, if deadenylation is inhibited in these cells to buffer against the lack of new mRNA synthesis, we would expect the steady-state tail length profile to be similar to that of G2 cells. Indeed, data from Park et al., 2016 (PMID: 27153541; Figure 1C) show that steady-state median poly(A)-tail lengths mostly unchanged between interphase and mitotically arrested cells. Therefore, in our study we opted to examine poly(A)-tail lengths in the presence of transcriptional inhibition, such that a direct comparison of deadenylation dynamics between G2 and M phases is possible. We have clarified this point in the main text.

In fact, I do not fully understand why the authors felt that poly(A) tail length in G2 versus M needed to be examined after transcriptional inhibition (Fig. 2B/C). Presumably to somehow get the kinetics. But M-phase arrested cells are already transcriptionally blocked (Fig. 1E). So this seems an unequal comparison. I'd appreciate it if the authors could explain their rationale.

As the reviewer notes, the goal of transcriptional inhibition in these experiments was to enable measurements of deadenylation dynamics rather than steady-state poly(A)-tail length distributions, as described in the point above. Mitotic cells are indeed transcriptionally inhibited even without addition of pharmacological transcriptional inhibitors. Thus, to directly examine the deadenylation behavior in both G2 and M samples over time under comparable conditions, we inhibited transcription in both sets of samples. Note that as with any drug treatment, actinomycin D could have unanticipated off-target effects. Therefore, we added the drug to both the G2 and mitotic samples to minimize differences between these two conditions. In addition, we conducted multiple experiments in mitotic cells without pharmacological transcriptional inhibition, which confirmed that our results hold true in mitotic cells even without actinomycin D treatment (Fig. 1, 3D). We have clarified this point in the main text.

The largest difference between G2 and M poly(A) profiles are the "phased PABPC toeprints" obvious in M-arrest but not G2-arrest (Fig. 3B/C). I think a deeper exploration and discussion of this difference would be useful. The authors give the impression that these toeprints are not normally observed and only seen after transcription inhibition (Eisen et al. 2020) or inactivation of CCR4-NOT (Yi et al., 2018). However, in my opinion, the Yi et al. paper already shows toeprints in control RNAi (Fig. 1F/G) and they are actually rather reduced in CCR4-NOT knock-down (unless I read their data wrongly). Furthermore, when Khalizeva et al. inhibited transcription in G2-arrested cells, this didn't make toeprints appear (Fig. 3B). So, it remains unclear to me what causes them. At least one transcript shows toeprints in G2 (Fig. EV4C). Can anything be learned from which transcripts do and which transcripts don't? Do any mRNAs in M-phase arrest not show toeprints?

We thank the reviewer for these comments. In analyzing the data from Yi et al. 2018 (PMID: 29932901), we do not observe clear toeprints for the bulk poly(A)-tail length distributions in the control (siNC) condition (as shown in Figure 1G of their paper). However, based on the data from Figures 3A&C and Figure 4 in Yi et al.'s paper, toeprints do appear following the knockdown of CCR4 (CNOT6). Thus, we believe that our data is consistent with prior work that toeprints are primarily observed under steady-state conditions when aspects of deadenylation are disrupted. We note that we do observe very subtle toeprints in G2 samples following 6 h of transcriptional inhibition, which is consistent with prolonged inhibition toeprint data from prior studies. Slobodin et al., 2020 (PMID: 32294471) found that prolonged inhibition of transcription leads to downregulation of CNOT6, thereby reducing deadenylation and stabilizing mRNA to buffer against transcriptome loss in the absence of novel mRNA synthesis. Although this study did not directly evaluate poly(A)-tail length distributions, these data in combination with the study by Eisen et al. (PMID: 31902669) support a model in which prolonged inhibition of transcription leads to inhibition of mRNA decay via repression of deadenylation in interphase cells. However, in mitotic samples the toeprints arise within as little as one hour following entry into mitosis, suggesting that an alternative mechanism must be responsible for rapid changes in deadenylation dynamics. We have clarified this point in the main text.

I also don't agree that the "striking phasing of PABPC toeprints.... suggest that PABPC is not efficiently removed from the poly(A) tails" in mitosis (page 8). It is entirely possible that PABPC is equally strongly bound in G2 and M, but that poly(A) tails have more flexible lengths beyond the PABPC-protected region in G2. It needs other types of measurements to demonstrate that there is less PABPC associated with G2 than with M transcripts or that PABPC is more strongly bound in M-phase arrested cells.

Overall, the most plausible interpretation is that some unidentified aspect of mRNA degradation is affected in M-phase arrested cells, and this may secondarily cause the phased poly(A) length profile. However, a global repression of deadenylation is not obvious in M- versus G2-arrested cells in my opinion. (Compare the G2/M difference here to how strongly poly(A) tail lengths shift when CCR4-NOT is knocked down in Yi et

al., 2018.) I feel the authors either need to provide stronger evidence or tone down their conclusion.

We agree with the reviewer that a higher inherent affinity of PABP for the poly(A)-tail during mitosis is not the only possible explanation for the toeprints. For instance, in our oligo(A) pulldown experiment for the revised paper, we did not observe differences in PABP association with oligo(A)-coated beads between mitotic and interphase lysates (Fig. 6A). It is also possible that the ability of CCR4–NOT to displace PABP may be inhibited in mitosis. For the revised paper, we evaluated whether the composition of the CCR4–NOT complex changes during mitosis (using a CNOT1 IP), but did not observe substantial changes (Fig. 6B). However, it is possible that the activity of this complex could be regulated in another way, such as post-translational modifications. Our model is based on our data as well as prior literature, which suggests that toeprints arise in response to downregulated deadenylation (see Slobodin et al., 2020 (PMID: 32294471) and Yi et al. 2018 (PMID: 29932901)).

We agree with the reviewer that the poly(A)-tail length distributions in our study do not reflect complete inhibition of the CCR4–NOT complex. This complex contains two different deadenylases, with CCR4 (CNOT6) capable of effectively evicting PABPC from the poly(A) tail, and CAF1 (CNOT7) primarily acting on unbound poly(A) tails (Yi et al. 2018 (PMID: 29932901)). Thus, we hypothesize that CCR4 (CNOT6) may have a reduced efficiency in mitosis compared to interphase, whereas CAF1 (CNOT7) may still be active. Consistent with this model, our new data show that CNOT6 tethering to an mRNA can induce transcript degradation in interphase, but not mitosis (Fig. 6C). This hypothesis is also consistent with the results in Yi et al. 2018, in which knockdown of each individual catalytic subunit (CCR4 or CAF1) did not have as strong of an effect on poly(A) tail length distributions as when these two perturbations were combined (Fig. 2B). Additionally, in Figures 3 and 4 of Yi et al.'s study, inhibition of CCR4 caused only a subtle shift in the distribution, with PABPC toeprints appearing.

Due to the distinctive mitotic PABP toeprints, as well as the robust restoration of mRNA decay in mitotic cells upon PABP depletion, we believe that it is reasonable to suggest that the persistent presence of PABP contributes to the stalling of deadenylation and is necessary for mRNA stabilization in mitosis. We have clarified this model in the main text.

(3) Similarly, I don't think it is justified to say that PABPC stabilizes transcripts specifically in mitosis (page 9). The PABC1&4-depletion experiment doesn't allow this conclusion. Yes, PABC1&4 depletion destabilizes mRNAs in the mitotic arrest. However, G2 was not tested. And Xiang and Bartel, eLife 2021, show that mRNA half-lives in asynchronously growing HeLa cells were reduced after PABPC knockdown. So whether this effect of PABPC on mRNA stability contributes to the enhanced mRNA stability in a mitotic arrest is unclear.

We agree with the reviewer that a key function of PABPC is to stabilize transcripts in a range of conditions, including during interphase. However, the fact that that PABPC1&4 co-depletion can “override” the inhibition of the mRNA degradation

pathway that occurs in mitotic cells (Fig. 2) provides critical information about which steps in the mRNA degradation pathway are attenuated. If steps downstream of deadenylation were completely inactivated, even PABPC1&4 depletion would not have resulted in a decrease in mRNA stability. We have clarified this logic in main the text, emphasizing the implications of the PABP depletion experiments for the state of mRNA degradation machinery in mitosis.

In the revised version of the manuscript, we have additionally provided data on possible contributions of deadenylase recruitment to mitotic stabilization (see Fig. 6C and main text). Using a strategy in which we tethered the key mRNA deadenylases CNOT6 or CNOT7 to a reporter transcript, we find that this induced targeting does not to override the mitotic stabilization of reporter mRNA. This means that these factors may be inactivated during mitosis via other mechanisms, such as post-translational modifications, preventing their activity regardless of their localization. In contrast, using siRNA treatment to induce the internal cleavage of a reporter mRNA, we demonstrate that endonuclease-mediated mRNA decay is still functional during an extended mitotic arrest (Fig. 3). This provides an elegant strategy for mitotic cells to preserve mRNAs in their most functional, translation-competent form, as opposed to decapped and deadenylated transcripts, which would be targeted for immediate degradation upon mitotic exit.

(4) The authors observe mitotic slippage in cells depleted of PABPC1&4 using the AID system. This experiment would benefit from a control where cells are treated with IAA without the presence of the AID tags to demonstrate specificity. In addition, since the authors want to conclude that destabilization of the mRNAs leads to slippage, can they show that slippage is rescued by blocking mRNA degradation (maybe by inhibiting decapping)? Without any such evidence, wouldn't it be possible that PABPC1&4 depletion impairs translation, and this leads to slippage?

We agree with the reviewer that it is formally possible that PABP depletion may have deadenylation-independent effects on translation. However, prior work found that PABPC1&4 depletion did not affect translation efficiency in this cell line, when depleted either using the AID system used in this paper or using siRNAs (Xiang et al. 2021, Figure 6). We have added this information to the main text.

(5) Although I appreciate the conciseness of the text, a little more information would occasionally be helpful. For example, the main text would benefit from information on the length of drug treatments or how cells were synchronized in G1 (release from mitotic arrest).

We thank the reviewer for this point. We have now provided additional information on the duration of the drug treatments and the G1 synchronization strategy to the main text and Figures of experimental schematics.

(6) I feel the prior literature could be more thoroughly and more accurately covered. My suggestions below are not comprehensive, just some examples. A surprising omission

in the references is Park Mol Cell 2016 (<https://doi.org/10.1016/j.molcel.2016.04.007>) where poly(A) tail lengths in S and M phase were measured. Other examples are: Novais-Cruz et al., eLife 2018 (<https://doi.org/10.7554/eLife.36898>), and Sloss et al., Oncotarget 2016 (<https://www.oncotarget.com/article/6894/>). Krenning et al., eLife 2022, is relevant and is indeed cited, but could be discussed more thoroughly. For example, they also show stable mRNAs in a mitotic arrest. On page 2/3 "Following the resolution of any mitotic damage, ...": Sinha et al., Cell Cycle 2019, does not seem an ideal citation here since it mostly deals with "slippage", i.e. not resolution, and is focused on drug-resistance in cancer.

We thank the reviewer for highlighting these omissions. We apologize for the oversight. We have now edited the text to include the relevant citations suggested by the reviewer and add more information from these studies.

(7) The Xiang and Bartel, eLife 2021 paper that is cited for PABC1&4 AID only reports PABPC1-AID. Please provide the missing information on PABPC4-AID.

We thank the reviewer for highlighting this omission. We apologize for the oversight. We have now edited the Methods section to include the relevant information. We have also added data demonstrating efficient depletion of PABPC4 upon IAA treatment (Fig. EV5A)

(8) I did not see any information on data availability.

The sequencing data is now available on Gene Expression Omnibus GSE314020 (RNA-seq) and GSE314021 (PAL-seq). We have also now added a statement on data availability to the main text.

(9) The authors occasionally pick single transcripts to give examples, e.g. MYC, KLF10, ATF3 in Fig. 2 or UBE2C in Fig. EV4. I wonder whether the data could be additionally mined for interesting, gene-specific information. For example: (1) The M/G2 half-life ratios for coding and non-coding RNAs have a distinct shoulder or second peak (Fig. 2F). What is the identity and nature of these RNAs with a considerably higher ratio?

The identities of the lncRNAs in the second peak are as follows: LINC00467, LINC01547, LINC00205, CYP4F26P (pseudogene), LINC00899, ZNF584-DT (divergent transcript), KANTR (protein-coding lincENSG00000232593), LINC00322, NEAT1, SCAMP1-AS1, ENSG00000249199 (novel gene), ENSG00000273760 (novel gene). Most of these are relatively lowly expressed, which may lead to higher estimates of mitotic half-life (artificially capped at 200 h; see Methods). The list of protein-coding mRNAs that are in the shoulder of the M-to-G2 half-life ratio plot contains over 1000 transcripts. This information is reported in the supplementary data tables.

(2) Clearly, at least one RNA shows distinct PABPC toeprints in G2 (Fig. EV4). Fig. 3B shows the average of thousands of RNAs. Which fraction of them shows toeprints? And what is the nature of the ones that do?

Indeed, we have identified at least one mRNA with distinct PABPC toeprints in G2 – *UBE2C*. Based on looking through the data, we suspect there are more examples of transcripts that behave similarly. However, we were unable to rigorously identify others or make confident statements about the fraction of such transcripts. Although this is an interesting question that could highlight unique regulation of deadenylation of individual transcripts in interphase cells, we believe that more detailed statistical analysis and modeling of these distributions is beyond the scope of the current paper.

Minor comments:

- What is the reason for the R^2 cutoff in the mRNA half-life experiments? The explanation given ("in order to allow for the possibility of global mRNA stabilization") is not sufficiently clear. What fraction of genes was removed? This information is important, as the filter may change the estimate of the median mRNA half-life across the genome.

The R^2 cutoff was imposed in interphase samples to filter out noisy data that compromised our confidence in the half-life calculations. Mitotic fits were not filtered due to a global lack of decrease in mRNA levels. The fraction of transcripts removed was <5% in the G2 vs M experiments (Fig. EV2B).

- PAL-seq experiment: How many replicates? Over 6 hours the median tail length in G2 decreased by 14 nt and in mitotic arrest by 9 nt. Because these numbers are fairly small, it would be useful confirm that they are not the result of random sampling effects with some statistical testing.

Due to the Illumina platform no longer providing users access to the intensity data from sequencing runs, we were only able to perform one replicate of PAL-seq, which limits our ability to perform statistical testing. We have added this information to the relevant Figure legends.

- Everything in this paper deals with a mitotic arrest situation, not with unperturbed mitosis. I think it is important to consistently use "mitotic arrest" phrasing in the conclusions and avoid giving the impression that the results apply to "mitosis" in general - although it is of course valid to speculate that they might.

We apologize for the oversight. We have now made out phrasing consistent throughout the paper.

- Maybe highlight the genes shown in Fig. 2C in the Fig. 2B and 2D graphs - just to illustrate where they fall on the spectrum.

We thank the reviewer for this suggestion. We have now highlighted the transcripts on the relevant plot.

- Why does 2E show the M/G1 ratio when M/G2 is shown in all other graphs? Show M/G2 in addition?

We chose to use the M/G1 ratio for this plot due to the fact that the TE data was collected from asynchronous cells, which in this cell line are predominantly in G1. Therefore, the M/G1 half-life ratio is the most relevant comparison to M/Async TE ratio.

- Abstract "... show that mRNA stabilization in mitosis is dependent on ... PABC1&4". While this statement is correct, it gives the impression that this is a mitosis-specific effect, which it very likely is not (also see above). If I am not mistaken, Xiang and Bartel, *eLife* 2021, showed a decreased mRNA half-life in asynchronously growing cells.

We have now provided a more thorough discussion of this point in the main text.

- Page 2, "downregulated transcription, as evidenced by chromosome condensation". This seems a little misleading since downregulated transcription does not require chromosome condensation (Zhao et al., *Nat Genetics* 2024, 10.1038/s41588-024-01759-x, or Spencer et al., *JCB* 2000, <https://doi.org/10.1083/jcb.150.1.13>)

We have changed the wording to reflect this: "downregulated transcription, accompanied by chromosome condensation"

- Page 4, "very close to their initial levels even after 24 hours of mitotic arrest". Please put a number on this. The graph is in log2 scale, which makes any changes look a little smaller than they are. Maybe report the median fold change between each time point. Similarly, on page 11 "maintained over a 24-hour mitotic arrest" isn't quite accurate, since mRNA levels go down to about 70 % after 24 hours.

We thank the reviewer for this comment. We have added the fold-change measurements to the main text. There was no change between 0 and 6 h, and a median 32% reduction in mRNA levels by 24 h.

- Page 5 "no correlation between the changes in mRNA half-life and translation efficiency" might be too strong for a correlation coefficient of -0.12 (small but not extremely small). "no strong correlation" would work better in my opinion.

We thank the reviewer for this comment. We have fixed our wording.

- Page 6, typo in "deadynylases".

We thank the reviewer for this comment. We have fixed the typo.

- Page 10 "range of fold-change values was also wider..., consistent with restored regulation..." I do not understand the authors' conclusion. Could they rephrase?

We thank the reviewer for this comment. We have added an explanation of our logic to the main text.

- Page 12 "leads to a strong reduction mRNA stability" should be "reduction in mRNA stability".

We thank the reviewer for this comment. We have fixed the typo.

- Page 12 "the effect of PABPC1&4 depletion... likely to involve ... stabilizing" Shouldn't this be "destabilizing" since the sentence talks about the AID experiment?

We thank the reviewer for this comment. We have fixed the typo.

- Page 15 "Cells were washed with 1mL cold PBS and permeabilized with 1mL of 3.7% formaldehyde" should probably read "fixed with 1 mL of 3.7 % formaldehyde..."

We thank the reviewer for this comment. We have fixed the typo.

- Methods paragraph on half-life calculations: "mRNA read counts were normalized to spike-in or mitochondrial RNA read counts in pharmacologically transcriptionally inhibited and uninhibited cells, respectively." Not quite correct. Fig 1 experiment does not use transcription inhibition but does use spike-ins.

We thank the reviewer for this comment. We did not calculate half-lives in the experiment in Figure 1 and therefore we did not include it in this section.

- Fig. 3 legend, re-phrase "Insets...".

We thank the reviewer for this comment. We have rephrased this sentence.

- Fig. 3B-C legend: should read "distributions in (B) G2- or (C) M-arrested cells..."

We thank the reviewer for this comment. We have fixed the error.

- Legend to Fig. EV3 uses both "G2- or M-arrested" and "RO-3306- or STLC-arrested". Maybe make the phrasing consistent.

We thank the reviewer for this comment. We have made our descriptions consistent.

- All the information from Fig. 4D is also obvious in Fig. 4C. What is the benefit of the additional plot?

We agree with the reviewer that these panels represent the same data. However, we included both representations, as the scatterplot highlights individual transcript differences, whereas the overlaid histograms highlight the general spread and shift of the data.

- *What do the error bars in Fig. EV1A and EV1B represent? If these are from the two replicates, rather show the individual points from the two replicates.*

We thank the reviewer for this comment. We have added individual points to Fig. EV1A. Figure EV1B contains both confidence intervals and the points from each of the two replicates.

- *What do the error bars in Fig. EV5C represent?*

We thank the reviewer for catching this omission. We have added this information to the figure legend.

Referee #3:

I was not entirely convinced that the authors have shown conclusively that reduced mRNA turnover was a consequence of a mitotic inhibition of deadenylation. The authors provide some measurements related to polyA tails and perform perturbations to the polyA binding proteins PABPC, but the evidence that deadenylation is reduced in mitosis is indirect at best, and, in my view, currently not strong enough to warrant the strong conclusion made in this study (even in the title); that altered deadenylation is the cause of altered mRNA stability in mitosis. If the authors can provide direct proof that deadenylation is inhibited during mitosis and that this inhibition causes the reduced mRNA turnover, then I think this paper would be well suited for publication.

I found the logic in the section describing figure 3 hard to follow. It wasn't clear to me how phasing of PABPC toeprints would represent proof of reduced deadenylation activity in mitosis.

We have now added a more detailed discussion of this point to the main text.

The authors deplete PABPC1/4 and observe that mRNA becomes less stable in mitosis and conclude that:

"This result suggests that the global transcriptome stabilization observed in mitotically arrested cells occurs through the inhibition of deadenylation."

I don't see how this conclusion can be reached from this experiment. PABPC1/4 are well known mRNA stabilizing protein, depleting them is expected to cause a decreased mRNA stability irrespective of the underlying mechanism that leads to reduced mRNA turnover in mitosis.

We agree with the reviewer that PABPC stabilizes transcripts across a range of conditions, including during interphase. However, we found it surprising and informative that PABPC1&4 depletion could “override” the inhibition of the mRNA degradation pathway that occurs in mitotic cells (Fig. 2). If deadenylation and downstream steps downstream were completely inactivated in mitotic cells, even PABPC1&4 depletion would not have resulted in a decrease in mRNA stability. We have clarified this logic in main the text, emphasizing the implications of the PABP depletion experiments for the state of mRNA degradation machinery in mitosis.

Moreover, if deadenylation was globally inhibited in mitosis, why are mRNAs rapidly degraded upon PABPC1/4 removal (as such decay likely involves deadenylation)? The experiment rather suggests that the deadenylation machinery is still active in mitosis (although this experiment doesn't really provide a quantitative analysis of deadenylation activity in mitosis vs interphase, it just shows that at least some deadenylation activity is present in mitosis).

We apologize for the miscommunication of our model here. Deadenylation by the CCR4–NOT complex is mediated by 2 nucleases, CNOT6 and CNOT7. Prior work from Narry Kim's lab (Yi et al., 2018) suggested that CNOT6 knockdown produces

toeprints, whereas CNOT7 knockdown does not. Given that CNOT6 knockdown matched our tail length profiles in mitosis, we hypothesized that CNOT6 may be inhibited during mitosis to give rise to the toeprints but CNOT7 is still active. Therefore, when PABP is depleted, CNOT7 can function to remove the free segments of the poly(A) tail. Notably, work from Narry Kim's lab showed that CNOT7 is not capable of displacing PABP, such that active CNOT7 without active CNOT6 should not induce mRNA degradation. We have clarified our reasoning on this in the main text.

The authors show that polyA tails are slightly longer in mitosis than in interphase (~10-20%), but it is not clear whether the difference in polyA tail length is large enough to explain the much larger difference in mRNA stability (3-4 fold). Moreover, no clear evidence is provided that show that the slight increase in polyA length in mitosis is in fact causative for the altered mRNA stability.

We agree with the reviewer that the inhibition of deadenylation may not be the only factor that contributes to the observed mRNA stabilization in mitotic cells. We have now updated these statements to reflect additional possibilities. We have also added new data exploring the possible contributions of downstream factors, such as XRN1 and the exosome following siRNA-mediated endonuclease cleavage (Fig. 3), as well as recruitment of deadenylation machinery (Fig. 6C). Also see reviewer #2, point (3).

Minor point:

In the introduction the authors cite a median mRNA half-life of 2-4 h, this seems on the short side, could the authors check additional literature references?

These median half-lives are indeed shorter than many published measurements. However, many of the measurements reported in the literature are obtained from transcriptionally inhibited cells (e.g., treated with actinomycin D, alpha-amanitin, and other drugs). These compounds have well-established side effects, which include mRNA stabilization. Therefore, measurements from transcription inhibition timecourses tend to overestimate the real mRNA half-lives (Lugowski et al., 2018; PMID: 29247756). With this in mind, we prioritized studies that obtained mRNA stability measurements using metabolic labeling assays. These are the studies that report median mRNA half-lives in the 2-4 h range.

Prof. Iain M Cheeseman
Whitehead Institute for Biomedical Research
Massachusetts Institute of Technology
Department of Biology
Suite 401, 455 Main St.
Cambridge, MA 02142

26th Feb 2026

Re: EMBOJ-2025-121943R-Q
Global stabilization of the transcriptome in mitotic cells

Dear Iain,

Thank you again for submitting a new version of your earlier manuscript on mitotic transcriptome stabilization for our consideration. I sent it back to the two referees that had initially raised substantive criticisms. As you will see from their comments below, both of them consider the study significantly improved and would now in principle support publication, pending additional presentational revisions to further clarify certain points and moderate certain conclusions and interpretations. I am therefore now inviting you to a formal round of minor revision in response to the remaining points, and for taking care of a number of editorial issues as follows:

- Regarding data presentation, I noticed that legends for Fig 4 refer to N=1 biological replicates, which sounds somewhat confusing, given that such statements are usually intended to indicate reproducibility and basis for statistical analysis, which may not apply for this sort of data? Please revisit and clarify.
- Related to that, several figures display means (and sometimes SEM) from only 2 biological replicates. Since this number is insufficient for meaningful statistics, please change all respective graphs to plotting individual data points (with their mean but no error bars) instead.
- Please adjust the order and the headers of the different manuscript sections: Title page with complete author information, Abstract, Introduction, Results, Discussion, Methods, Data Availability, Acknowledgements, Disclosure and Competing Interests Statement, References, Main Figure Legends, Tables, Expanded Figure Legends.
- Please remove the Highlights section from the manuscript, and instead upload them separately, prefaced by a short 'blurb' text summing up the conceptual aspect (background and main thrust) of the study in two sentences (max. 250 characters), followed by the highlights i.e. brief factual statements of key results of the paper; they will form the basis of an editor-written 'Synopsis' accompanying the online version of the article. Please also upload a synopsis image, which can be used as a "visual title" for the synopsis section of your paper. The image should be ideally in JPG format, and please make sure that it remains in the modest dimensions of (exactly) 550 pixels wide and between 300-500 pixels high.
- Please carefully go through the reference list and double-check that all references have their complete citation information - including year, journal name & volume, and page/locator numbers.
- In the "Data Availability" section, please include a direct access link to the database(s) in which datasets derived in this work, and referenced with their accession codes, have been deposited.
- Please double-check to make sure to all relevant funding information in the manuscript is congruent with the info entered into our submission system. Currently missing in the submission system are:
MIT HEALS graduate fellowship, NIH fellowship (K00CA234921), Natural Sciences and Engineering Research Council of Canada
They need to be added to the submission system as separate entries (via 'More Funders' option).
- Please include a separately-uploaded Reagents and Tools table, adhering to the template table downloadable from our author guidelines/linked below: <https://www.embopress.org/page/journal/14693178/authorguide#structuredmethods>
- Please double-check all Figure call-outs in the text (a callout for Fig 7G seems to be missing, and some figures appear to be wrongly referenced as 'Supp Fig 2B/C').
- Please rename the 2 supplemental tables into Expanded View Tables (naming/referencing: 'Table EV1/2'), and upload them both as separate DOC or XLS files.

- Please rename the 4 supplemental data files as Expanded View Datasets (naming/referencing: 'Dataset EV1-4'), and include a separate 'Legends' tab in each respective XLSX file.

- Finally, during routine pre-acceptance checks, our data editors have raised the following queries regarding figures, data, and legends; I would appreciate if you briefly answered to them in the cover letter of your final submission, and made the requested text modifications with changes/additions highlighted via the "Track changes" option, to facilitate our final checking:

1. Please note that the box plots need to be defined in terms of minima, maxima, centre, bounds of box and whiskers, and percentile in the legends of figures 1B, EV2 B
2. Please note that information related to n is missing in the legends of figures EV2 B
3. Please define the annotated p values ****/****/**/* as well as provide the exact p-values for the same in the legend of figure 7F as appropriate.
4. Please supply source data for blots in Figure 7A and Figure EV5A, and consider including a better contrasted version of the former. Related to that, you shall also receive a separate message from our Source Data curation team, with instructions on how to prepare and upload relevant image and numerical raw data.

I am returning the manuscript to you for a final round of minor revision, to allow you to make these modifications and upload the revised files. Once we will have received them, we should hopefully be able to swiftly proceed with formal acceptance and production of the manuscript.

With kind regards,

Hartmut

*** PLEASE NOTE: All revised manuscript are subject to initial checks for completeness and adherence to our formatting guidelines. Revisions may be returned to the authors and delayed in their editorial re-evaluation if they fail to comply to the following requirements. As a first step please read our guidelines for revised submissions:
<https://link.springer.com/journal/44318/submission-guidelines#cms-Revised-submissions>

1) Every manuscript requires a Data Availability section (even if only stating that no deposited datasets are included). Primary datasets or computer code produced in the current study have to be deposited in appropriate public repositories prior to resubmission, and reviewer access details provided in case that public access is not yet allowed.

4) Each main and each Expanded View (EV) figure should be uploaded as individual production-quality files (preferably in .eps, .tif, .jpg formats). For suggestions on figure preparation/layout, please refer to our Figure Preparation Guidelines:
<https://media.springernature.com/original/springer-cms/rest/v1/content/27825798/data/v1>

6) Please complete our Author Checklist, and make sure that information entered into the checklist is also reflected in the manuscript; the checklist will be available to readers as part of the Review Process File.

8) Please note that supplementary information at EMBO Press has been superseded by the 'Expanded View' for inclusion of additional figures, tables, movies or datasets; with up to five EV Figures being typeset and directly accessible in the HTML

version of the article.

9) To facilitate reproducibility and cross-laboratory adoption of methodologies, please structure the Materials & Methods section as outlined in our guide to authors, including a completed Reagents and Tools Table.

10) Digital image enhancement is acceptable practice, as long as it accurately represents the original data and conforms to community standards. If a figure has been subjected to significant electronic manipulation, this must be clearly noted in the figure legend and/or the 'Materials and Methods' section. The editors reserve the right to request original versions of figures and the original images that were used to assemble the figure. Finally, we generally encourage uploading of numerical as well as gel/blot image source data.

In the interest of ensuring the conceptual advance provided by the work, we recommend submitting a revision within 3 months (27th May 2026). Please discuss the revision progress ahead of this time with the editor if you require more time to complete the revisions. Use the link below to submit your revision:

Link Not Available

Referee #2:

Khalizeva, Ly, Cheeseman, and colleagues have addressed how cells arrested in mitosis maintain their mRNA pool despite the absence of measurable transcription. This is a fundamental question that may have relevance in cancer biology. The authors demonstrate that mRNA half-life is globally reduced, which is a novel finding. Although the exact mechanism remains unclear, additional experiments narrow down how the regulation may work.

The authors have adequately addressed all my previous comments.

The new siRNA (Fig. 3) and tethering data (Fig. 6) add valuable information. I also appreciate the rephrasing of the rationale for the PABC1&4 depletion experiment.

I have a few minor points that the authors may address as they see fit:

- From the experiment shown in Fig. 6A (oligo(A) pull down), the authors conclude that there were no differences in oligo(A) association between interphase and mitotic cells. However, there are points off-axis in these data (in particular, some proteins enriched in the oligo(A) pull-down in asynchronous, but not mitotic cells). What are the statistics or rationale behind the conclusion that there are no differences?
- The siRNA experiment in Fig. 3 is a glass half-full/half-empty situation. Clearly, some degradation still occurs in M-phase cells. But, it seems to be less than in G2-phase cells. (A statistical test may be appropriate.) The authors chose to emphasize that degradation still occurs. Maybe adding a half-sentence to indicate the difference would make this clearer ("... levels ... were reduced, although not to the extent observed in G2 cells").
- A very minor semantic point:
When talking about the role of PABC1&4 in mitotic RNA stability, the authors use "stabilization" (e.g., "mRNA stabilization ... is dependent on PABC1 and PABC4", "PABC1&4 are necessary for mitotic mRNA stabilization"). I would prefer the word "stability", since PABC1&4 certainly are required for keeping these mRNAs stable, but it may not be the element that is specifically regulated to make the difference between interphase and mitosis.
- On page 12, regarding Fig. 7D, "the range of fold-change values was ... wider in PABP-depleted mitotic cells ... consistent with restored regulation of mRNA stability to achieve transcript-specific degradation rates". I feel additional evidence is required to conclude that transcript-specific degradation rates are achieved. The authors would have the transcript-specific degradation rates from interphase available to do that.
- The citations for "ongoing translation is essential for cell survival" or "ongoing translation is necessary for the maintenance of mitotic arrest" could include additional earlier evidence (e.g., Taylor lab).

Other:

- Figure legends for Fig. 5 and 6 are swapped.
- There is a legend for Fig. EV2G, but no figure panel.

Referee #3:

In this revised manuscript the authors have provide new insights that support the idea that deadenylation is decreased in mitosis, resulting in decreased mRNA turnover. I found the experiments in which CNOT6 is tethered to the mRNA especially compelling. Some of the other new experiments are, however, less convincing and I think the description of these results would benefit from a more nuanced interpretation, see below.

The authors conclude from their new siRNA experiments that the exonucleolytic decay pathways are not inhibited in mitosis. I don't see how they can come to this conclusion from figures 3B and 3C. There is a clear difference in mRNA turnover in G2 and mitosis of both cleavage fragments. Wouldn't one interpret these results as there being an reduction in exonucleolytic decay in mitosis?

It should also be noted that the authors don't assess whether siRNA cleavage efficiency is similar in G2 and mitosis, which could impact the outcome of this experiment. It is possible that the difference in fragment levels are solely due to differences in cleavage efficiency, or alternatively that the exonucleolytic pathways are reduced in mitosis. This seems like an important point for this manuscript.

The authors perform an oligoA pull down in either interphase or mitosis and show that many interactors are shared in both cell cycle phases, concluding that differential interactome of the polyA tail is likely not responsible for differences in mRNA stability. I certainly agree that many factors are found as interactors in both cell cycle phases, but I think it goes to far to state that there are no differences. Several proteins show >2 fold difference in interaction in G2 vs mitosis, and the authors cannot really conclude based on the current data whether such differences are functionally relevant.

We thank the reviewers and the editor for their positive assessment of our resubmission and for the constructive feedback on the updated manuscript. We have edited the manuscript to address these comments. Please find our point-by-point response to reviewer comments below.

Referee #2:

1. From the experiment shown in Fig. 6A (oligo(A) pull down), the authors conclude that there were no differences in oligo(A) association between interphase and mitotic cells. However, there are points off-axis in these data (in particular, some proteins enriched in the oligo(A) pull-down in asynchronous, but not mitotic cells). What are the statistics or rationale behind the conclusion that there are no differences?

In this experiment, the differentially enriched proteins were examined for pathway relevance. Due to the in vitro nature of this experiment, we expected there to be a number of differentially associated proteins that would not reflect in-cell binding. Most of the proteins that showed differential enrichment appeared to be artifacts or noise. For example, PIGR (Polymeric immunoglobulin receptor) and LYZ (lysozyme C) were two of the most differentially pulled-down proteins. As these are not on pathway, we discounted these proteins in our interpretation of the results, but nonetheless chose to include them in the Figure and the accompanying dataset in case future work uncovers relevant hits we may have missed. We have added a concise explanation of these off-the-diagonal points to the main text of the manuscript.

2. The siRNA experiment in Fig. 3 is a glass half-full/half-empty situation. Clearly, some degradation still occurs in M-phase cells. But, it seems to be less than in G2-phase cells. (A statistical test may be appropriate.) The authors chose to emphasize that degradation still occurs. Maybe adding a half-sentence to indicate the difference would make this clearer ("... levels ... were reduced, although not to the extent observed in G2 cells").

We have added a clause to more carefully describe the result, as the reviewer suggested.

3. A very minor semantic point: When talking about the role of PABC1&4 in mitotic RNA stability, the authors use "stabilization" (e.g., "mRNA stabilization ... is dependent on PABC1 and PABC4", "PABC1&4 are necessary for mitotic mRNA stabilization"). I would prefer the word "stability", since PABC1&4 certainly are required for keeping these mRNAs stable, but it may not be the element that is specifically regulated to make the difference between interphase and mitosis.

We appreciate the thought the reviewer put into this phrasing. Our data show that mitotic stabilization of the transcriptome does not happen without PABPC1&4 present (Fig. 7C, D). Therefore, we think it is reasonable to interpret these results as an indication of the role of PABPC1&4 in mitotic mRNA stabilization.

4. On page 12, regarding Fig. 7D, "the range of fold-change values was ... wider in PABP-depleted mitotic cells ... consistent with restored regulation of mRNA stability to achieve transcript-specific degradation rates". I feel additional evidence is required to conclude that transcript-specific degradation rates are achieved. The authors would have the transcript-specific degradation rates from interphase available to do that.

We have removed the sentence.

5. The citations for "ongoing translation is essential for cell survival" or "ongoing translation is necessary for the maintenance of mitotic arrest" could include additional earlier evidence (e.g., Taylor lab).

We have now provided additional references from earlier studies, as the reviewer suggested.

6. Figure legends for Fig. 5 and 6 are swapped.

We thank the reviewer for catching this mistake. We have fixed the error.

7. There is a legend for Fig. EV2G, but no figure panel.

We thank the reviewer for catching this mistake. We have removed the extra legend.

Referee #3:

1. The authors conclude from their new siRNA experiments that the exonucleolytic decay pathways are not inhibited in mitosis. I don't see how they can come to this conclusion from figures 3B and 3C. There is a clear difference in mRNA turnover in G2 and mitosis of both cleavage fragments. Wouldn't one interpret these results as there being an reduction in exonucleolytic decay in mitosis?

We have added a qualification of this finding, as described above in response to reviewer #2, point 2.

2. It should also be noted that the authors don't assess whether siRNA cleavage efficiency is similar in G2 and mitosis, which could impact the outcome of this experiment. It is possible that the difference in fragment levels are solely due to differences in cleavage efficiency, or alternatively that the exonucleolytic pathways are reduced in mitosis. This seems like an important point for this manuscript.

We have now updated the interpretation of the siRNA experiments in terms of both endo- and exonuclease activity.

3. The authors perform an oligoA pull down in either interphase or mitosis and show that many interactors are shared in both cell cycle phases, concluding that differential interactome of the polyA tail is likely not responsible for differences in mRNA stability. I certainly agree that many factors are found as interactors in both cell cycle phases, but I think it goes to far to state that there are no differences. Several proteins show >2 fold difference in interaction in G2 vs mitosis, and the authors cannot really conclude based on the current data whether such differences are functionally relevant.

Please see response to reviewer #2, point 1.